# The AT-hook is an evolutionarily conserved auto-regulatory domain of SWI/SNF required for cell lineage priming

Dhurjhoti Saha[1,2], Solomon Hailu[1,2,3], Arjan Hada[1,2], Junwoo Lee [1,2], Jie Luo[4], Jeff A. Ranish [4], Yuan-chi Lin[1,2,5], Kyle Feola[1,6], Jim Persinger[1,2], Abhinav Jain [1,2], Bin Liu[1], Yue Lu[1], Payel Sen [7] & Blaine Bartholomew [1,2] ✉

The SWI/SNF ATP-dependent chromatin remodeler is a master regulator of the epigenome, controlling pluripotency and differentiation. Towards the C-terminus of the catalytic subunit of SWI/SNF is a motif called the AT-hook that is evolutionary conserved. The AT-hook is present in many chromatin modifiers and generally thought to help anchor them to DNA. We observe however that the AT-hook regulates the intrinsic DNA-stimulated ATPase activity aside from promoting SWI/SNF recruitment to DNA or nucleosomes by increasing the reaction velocity a factor of 13 with no accompanying change in substrate affinity ($K_M$). The changes in ATP hydrolysis causes an equivalent change in nucleosome movement, confirming they are tightly coupled. The catalytic subunit's AT-hook is required in vivo for SWI/SNF remodeling activity in yeast and mouse embryonic stem cells. The AT-hook in SWI/SNF is required for transcription regulation and activation of stage-specific enhancers critical in cell lineage priming. Similarly, growth assays suggest the AT-hook is required in yeast SWI/SNF for activation of genes involved in amino acid bio-synthesis and metabolizing ethanol. Our findings highlight the importance of studying SWI/SNF attenuation versus eliminating the catalytic subunit or completely shutting down its enzymatic activity.

ATP-dependent chromatin remodelers are critical regulators in the cell involved in transcription, DNA replication/repair and genome integrity and as such are also highly regulated themselves[1]. These remodeling complexes have been shown to have the potential to be auto-regulated, especially the ISWI family of remodelers, but less has been revealed about the auto-regulation of SWI/SNF[2]. In all SWI/SNF complexes, the AT-hook is located between the SnAC and bromodomains at the C-terminus of the catalytic subunit. The AT-hook motif is found in a number of other chromatin modifiers and has been suggested to

anchor these proteins to chromatin[3,4]. The AT-hook motif is a tripeptide of arginine-glycine-arginine flanked on either one or both sides by proline and was first identified in HMGA1 (High mobility group AT-hook 1). The AT-hook binds to the minor groove of DNA, alters DNA architecture, and facilitates the binding of other proteins[5–7]. Sequence alignment of a large number of AT-hook-containing proteins has revealed a 10-15 amino acid motif that has a G-R-P core surrounded by K and R residues[4,8,9]. A special class of AT-hook motifs called extended AT-hooks are approximately 3 times longer than the conventional AT-

[1]Department of Epigenetics and Molecular Carcinogenesis, Univ. of Texas MD Anderson Cancer Center, Houston, TX 77230, USA. [2]University of Texas MD Anderson Cancer Center, Center for Cancer Epigenetics, Houston, TX 77230, USA. [3]Illumina, 5200 Illumina Way, San Diego, CA 92122, USA. [4]Institute for Systems Biology, Seattle, WA 98109, USA. [5]BioAgilytix, Durham, NC 27713, USA. [6]Department of Internal Medicine (Nephrology) and Pharmacology, University of Texas Southwestern Medical Center, Dallas, TX 75390, USA. [7]Laboratory of Genetics and Genomics, National Institute on Aging, Baltimore, MD 21224, USA. ✉e-mail: bbartholomew@mdanderson.org

hook with basic residues extending symmetrically 12-15 amino acids from the core AT-hook motif and have an order of magnitude higher affinity for RNA than DNA[4,10]. The most characterized ATP-dependent chromatin remodeling complex in terms of the AT-hook is the nucleolar remodeling complex (NoRC) with Snf2h, a Snf2-like ATPase subunit, and Tip5 (TTF-I interacting protein 5), a regulatory subunit containing 4 AT-hooks and a TAM (Tip5/ARBP/MBD) domain[11,12]. The NoRC complex represses rRNA transcription by positioning nucleosomes at the promoters of rRNA genes and facilitating the association of rRNA genes with the nuclear matrix. The AT-hook within the Tip5 subunit of the NoRC complex is required for NoRC binding to rRNA gene promoters[13]. The AT-hooks in the auxiliary NURF 301 subunit[14] and Rsc1 and 2 subunits in yeast RSC complex[15] are important for efficient remodeling, but the mechanistic basis of these AT-hooks is not understood. Even though the AT-hook in the catalytic subunit of the SWI/SNF complex is conserved throughout eukaryotes, it is not known if it is required for nucleosome remodeling.

We find the AT-hook in the catalytic subunit of yeast SWI/SNF positively regulates remodeling by modulating the intrinsic DNA-dependent ATPase activity of the catalytic subunit without impacting SWI/SNF's affinity for DNA. The AT-hook is evolutionary important for SWI/SNF regulation as shown by being required for cell lineage priming in mouse embryonic stem cells and for regulating genes involved in amino acid biosynthesis in yeast. In embryonic stem cells, two forms of the SWI/SNF complex, esBAF and GBAF (or ncBAF) share the same Smarca4/Brg1 catalytic subunit, which we will refer to as Brg1, and are respectively localized at enhancers and promoters[16]. Brg1 is important for stem cell maintenance because it regulates the expression of NANOG, binds to the regulatory regions of the pluripotent genes including *Oct4* and *Sox2,* and regulates their expression in a feedback manner[17–21]. In the case of Oct4 and FoxD3, Brg1 association enhances the ability of these pioneer TFs to make DNA accessible at their binding sites for gene activation[20]. We now observe Brg1 and its AT-hook are required to promote binding of epiblast-specific TFs, crucial in early cell lineage priming. Besides TFs promoting SWI/SNF recruitment, the p300 and MLL3/4 co-activators also reciprocally facilitate in SWI/SNF binding, which in turn promotes H3K27 acetylation and H3K4 mono-methylation (H3K4me1)[22–24]. We find the acquisition of epiblast-specific patterns of H3K4me1 at enhancers, a key step in early cell lineage priming, requires Brg1 and its AT-hook. Although Brg1 is known to be important for stem cell maintenance, differentiation of cardiomyocytes, neural progenitor cells, and myotubules formation, we uncover for the first time its importance in early cell lineage priming.

## Results

### AT-hooks positively regulate the nucleosome mobilization activity of SWI/SNF and are not essential for complex integrity

Two AT-hooks are located at the C-terminus of yeast Snf2 between the Snf2 ATPase Coupling (SnAC) domain and bromodomain (Fig. 1a, b). First, we find the AT-hooks are not required for complex integrity by deleting both AT-hooks (ΔAT) from residue 1446 to 1530 and purifying the mutant complex by M2 agarose immunoaffinity chromatography and analyzing the purified complex by SDS-PAGE (Fig. 1c). We examine the nucleosome remodeling activity of WT and ΔAT SWI/SNF using nucleosomes assembled with the 601-nucleosome positioning sequence and 29 and 59 bp of flanking DNA (29N59) and recombinant *Xenopus laevis* histones. In order to examine a single round of SWI/SNF remodeling, a fixed amount of nucleosomes is titrated with SWI/SNF to determine how much is needed to saturate the nucleosomes (Fig. 1d). Using saturating SWI/SNF, we find ΔAT SWI/SNF mobilizes nucleosomes 13 times slower than WT SWI/SNF (0.26 nM/s for WT compared to 0.0019 nM/s for ΔAT), which is similar to the decrease in the rate of ATP hydrolyzed and therefore indicates the reduction in remodeling is tightly coupled to the decrease in ATPase activity (Fig. 1e–f). Given nucleosomes are fully bound by WT and ΔAT SWI/SNF, the differences in remodeling are not due to any reduction in the binding of ΔAT SWI/SNF.

### AT-hooks positively regulate the DNA- and nucleosome-stimulated ATPase activity of SWI/SNF

Coupling of the rate of ATP hydrolysis and nucleosome mobilization in ΔAT SWI/SNF prompted us to test the role of the AT-hook in regulating the DNA- and nucleosome-dependent ATPase activity of SWI/SNF by determining the velocity or reaction rate ($V_{max}$) and binding affinity for substrate ($K_M$) using the Michaelis-Menten approach for WT and ΔAT SWI/SNF (Fig. 2a–d and Table 1). Loss of the AT-hook does not impact the substrate affinity of SWI/SNF but does decrease the reaction rate by a factor of 13 with free DNA. The 3-fold increase of substrate affinity of SWI/SNF with nucleosomes compared to free DNA (98 versus 311 nM) was eliminated when the AT-hook was deleted from the complex (364 vs 340 nM, see Table 1). The rate of ATP hydrolysis is decreased 14-fold with ΔAT versus WT SWI/SNF in the presence of nucleosomes.

### The AT-hooks moderately contribute to the affinity of yeast SWI/SNF for nucleosomes

The affinity of ΔAT and WT SWI/SNF for DNA and nucleosomes was measured by electrophoretic mobility shift assays (EMSA) to directly assess the AT-hook contribution to SWI/SNF's affinity for free DNA and nucleosomes. The apparent $K_D$ (Dissociation Constant) of ΔAT was increased 1.6-fold more than WT for free 601 DNA (12.8 versus 7.87 nM,) and increased 2.7-fold for nucleosomes for ΔAT compared to WT SWI/SNF (20.5 versus 7.67 nM) (Fig. 3a, b and S1a, b). These data suggest the AT-hook has more of a role in promoting the affinity SWI/SNF for nucleosomes than for DNA consistent with that observed in the Michaelis-Menten data. We examined the ability of plasmid DNA to compete for SWI/SNF binding to either 601 DNA or nucleosomes. The Ki for competing SWI/SNF from DNA is ~2-fold lower for WT than ΔAT SWI/SNF (12 versus 25 nM), indicating the AT-hook promotes competition of plasmid DNA for binding SWI/SNF (Supplementary Fig. 2a, b). In contrast there is no difference in the estimated Ki observed for WT and ΔAT SWI/SNF when bound to nucleosomes which indicates the AT-hook does not promote the transfer of SWI/SNF from nucleosomes to free DNA and likely behaves differently when SWI/SNF is bound to nucleosomes.

Next, SWI/SNF recruitment to nucleosomes by the VP-16 activation domain fused to the Gal4 DNA binding domain (Gal4-VP16) did not require AT-hooks as seen by EMSA (Supplementary Fig. 1c). SWI/SNF binding to nucleosomes under these conditions is highly dependent on Gal4-VP16 binding to extranucleosomal DNA 29 bp from the edge of nucleosomes (Supplementary Fig. 1c compare lanes 3-5 to 6–8). The ΔAT complexes are recruited the same as wild-type SWI/SNF under these conditions and AT-hooks are not required for efficient recruitment of SWI/SNF by an acidic transcription factor (Supplementary Fig. 1c lanes 9–20).

### The AT-hooks of yeast Snf2 are in close proximity to the N-terminal tail of histone H3

To understand how SWI/SNF interacts with nucleosomes and the role of the AT-hooks in these interactions, we mapped protein-protein interactions (PPIs) in nucleosome-bound SWI/SNF by crosslinking-mass spectrometry (CX-MS) using the amine-reactive crosslinker bis[sulfosuccinimidyl] suberate (BS3). There is good concordance of the CX-MS data with the cryo-EM structure of nucleosome-bound SWI/SNF as seen by the majority of the lysine pairs observed in the intra- and inter-links having a Cα-Cα distance of ≤30 angstroms in the solved structure and are similar to that observed earlier for CX-MS of free SWI/SNF[25] (Supplementary Fig. 3). The Snf2, Snf5, Arp7, Swi1 and Snf6 subunits crosslinked to histones, consistent with the recent cryo-EM structures of nucleosome-bound ySWI/SNF[25,26] (Supplementary Fig. 3a). The histones crosslink the C-lobe, the linker region between

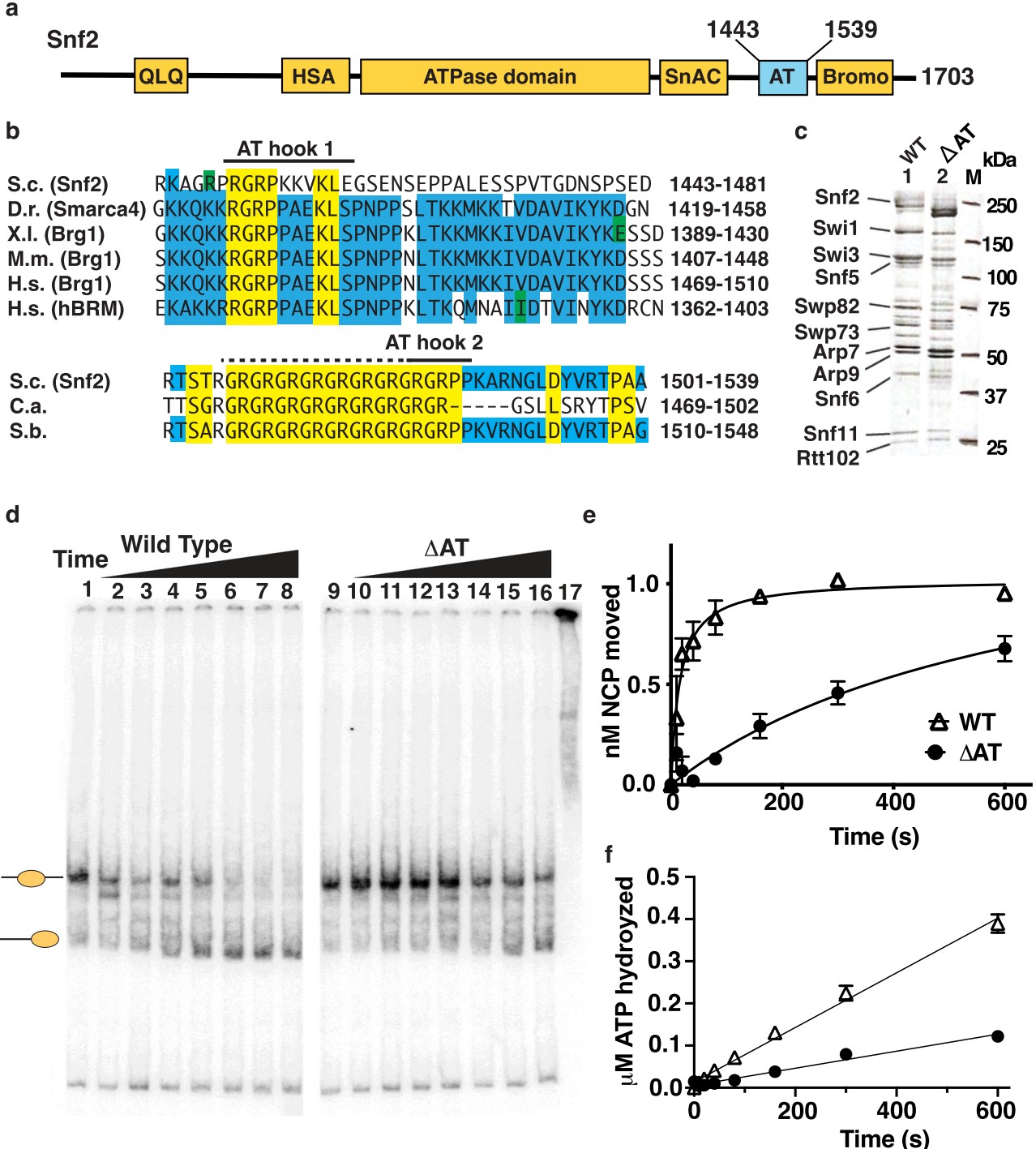

**Fig. 1 | The AT-hook of Snf2 is required for efficient ATP hydrolysis and nucleosome mobilization by the yeast SWI/SNF complex. a** The domain organization of the catalytic subunit of yeast Snf2 is shown including the two AT hook motifs at the C-terminus. The two AT-hooks of yeast Snf2 are removed by deleting residues 1443-1539 in yeast *SNF2*. **b** The amino acid sequence is shown for the two AT-hooks in Snf2 of *S. cerevisiae* and its homologs in zebra fish (D.r.), Xenopus laevis (X.l.), mouse (M.m), human (H.s.), and two fungal species *Candida albicans* (C.a.) and *Saccharomyces bayanus* (S.b.). Conserved residues are highlighted in blue and yellow. **c** Wild (WT) and the AT-hooks deletion mutant (ΔAT) SWI/SNF complexes were immunoaffinity purified and analyzed on a 4-20% gradient SDS-PAGE. **d** The nucleosome mobilizing activity of WT and ΔAT SWI/SNF are compared using an electrophoretic mobility shift assay (EMSA) on a 5% native polyacrylamide gel. The reactions contained 2.5 nM 29N59 nucleosomes, 7.5 nM yeast SWI/SNF and 4.4 μM ATP and incubated for 0, 10, 20, 40, 80,160, 300 and 600 s at 30 °C (WT lanes 1-8 and ΔAT lanes 9-16). Nucleosomes were fully bound by SWI/SNF under these conditions as shown for ΔAT SWI/SNF in lane 17. **e** The extent of remodeling was quantitated and plotted relative to reaction time in reactions that contained 7.5 nM SWI/SNF, 2.5 nM nucleosomes and 4.4 μM ATP. **f** The rate of ATP hydrolyzed under the same conditions as in (**e**) were measured by thin layer chromatography and plotted with the amount of ATP hydrolyzed versus reaction time. Two or three replicates were performed for both (**e**) and (**d**) and individual data points are shown. Source data are provided as a Source Data File.

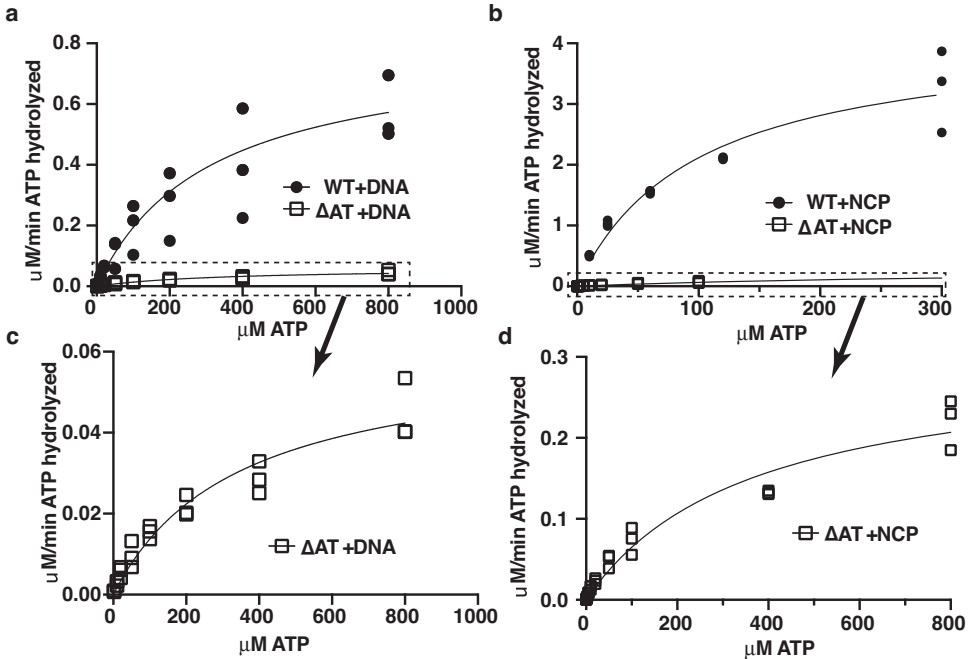

**Fig. 2 | The AT-hooks enhance the reaction velocity of ATP hydrolysis by SWI/SNF in the presence of DNA or nucleosomes. a, b** The rate of ATP hydrolysis was measured for SWI/SNF (1.6 nM SWI/SNF) with either (**a**) free DNA (35 ng pUC18 plasmid DNA) or (**b**) 29N59 nucleosomes (8 nM) with ATP that varied from 0.2 to 800 mM to determine the maximum velocity ($V_{max}$) and substrate affinity ($K_M$) for both wild type and ΔAT mutant SWI/SNF. These data are used in Table 1 to determine the $K_M$ and Vmax. **c, d** The data for ΔAT mutant SWI/SNF is shown with a different y-scale to better visualize the trend not readily seen in (**a, b**). Three replicates are performed and individual data points are shown. Source data are provided as a Source Data File.

the two lobes of the ATPase domain and the HSA domain in full agreement with the cryo-EM structure. We observe the C-lobe of the ATPase domain is in close proximity to the N-terminal tail of histone H4 by crosslinking to lysines 9, 13, and 21 of H4, consistent with that observed in the Snf2-nucleosome cryo-EM structure but not in the SWI/SNF-nucleosome structure[27,28] (Fig. 3c). These data demonstrate the H4 N-terminal tail may influence SWI/SNF remodeling in a manner similar to that previously observed for the ISWI family of chromatin remodelers[29–31] including interacting with the C-lobe of the ATPase domain[32]. The HSA domain associates with the N-terminal tail of H3, which agrees with the HSA domain and the H3 tail both contacting extranucleosomal DNA[33,34] (Fig. 3c).

The CX-MS data potentially reveals novel associations not previously observed in the cryo-EM structure regarding the AT-hook and SnAC domain. We find the C-terminus of the SnAC domain crosslinks to the C-terminus of H2B at lysines 117 and 121 and the AT-hooks crosslinks to the N-terminal tail of histones H3 at lysines 15 and 28 of H3. The close proximity of the AT-hook to the H3 tail is consistent with acetylation of H3 histone tail and the AT-hook positively regulating SWI/SNF recruitment and remodeling activity, similarly, ubiquitination of the C-terminus of H2B interfering with SWI/SNF remodeling is also congruent with SnAC binding at this site[27,35–38] (Fig. 3c). The SnAC domain is in close proximity to the C-lobe of the ATPase domain as seen by lysines 1320 and 1336 in SnAC crosslinking to lysines 1028, 1040 and 1041 in the N-terminus of the C-lobe when SWI/SNF is bound

to nucleosomes (Fig. 3d). The close proximity of the SnAC domain with the C-lobe of the ATPase domain are consistent with biochemical data showing the SnAC domain is a positive regulator of the ATPase activity of SWI/SNF[25,26,39]. The AT-hook and SnAC domain are also in close proximity to each other as observed by lysine 1314 in the SnAC domain crosslinking lysine 1441 in the first AT-hook (Fig. 3e).

**Disruption of the AT hooks alters the in vivo activity of SWI/SNF**
The impact of loss of the AT-hooks on the in vivo activity of SWI/SNF is assessed by examining cell growth under conditions where the SWI/SNF complex is required for viability such as amino acid deprivation or switching metabolism to alternative carbon sources. Amino acid starvation regulates synthesis of the Gcn4 TF and Gcn4 activates the transcription of about 200 genes including most of the amino acid biosynthetic genes[40]. Gcn4 helps recruit SWI/SNF independent of the histone acetyltransferase Gcn5, the catalytic subunit of the SAGA complex, and SWI/SNF is required for activation of many of these genes[41,42]. The AT-hook of Snf2 is required for SWI/SNF to rescue cells from amino acid starvation induced by the addition of sulfometuron methyl (SM), similar to when the SnAC domain is deleted (Fig. 4). However, loss of Snf2 (*snf* 2Δ) or the catalytically dead version of Snf2 (*snf* 2-K798A) has a more dire effect on cell viability with SM than deletion of the AT-hook or SnAC domain, consistent with the degree in which SWI/SNF remodeling is likely lost. The ATPase activity of the catalytically dead Snf2 mutant has previously been shown to be at least an order of magnitude more inhibited than the AT hook or SnAC domain deletion mutant complexes[43]. The effects of loss of the AT-hook appears to likely be independent of the bromodomain, given that loss of the bromodomain was previously shown to have no phenotype in the presence of SM[38]. In the switch from glucose to raffinose, deletion of the AT-hook or SnAC domain has no observable effect while cell growth is severely inhibited for both the *snf* 2Δ and *snf* 2-K798A strains (Fig. 4). Cell growth in the presence of ethanol in place of glucose largely mirrors that observed with SM in that the AT-hook and SnAC deletion are not as deleterious as the loss of *SNF*2 or its more complete

**Table 1 | Summary of binding affinity between WT and dAT yeast SWI/SNF**

|  | Km (nm) | 95% CI of Km | Vmax (μM/min) | 95% CI of Vmax |
|---|---|---|---|---|
| WT + DNA | 311 | 188–541 | 0.795 | 0.664 to 1.04 |
| dAT + DNA | 340 | 206–593 | 0.0604 | 0.0489 to 0.0792 |
| WT + NCP | 98 | 55.3–178 | 4.19 | 3.36 to 5.47 |
| dAT + NCP | 364 | 218–681 | 0.301 | 0.246–0.405 |

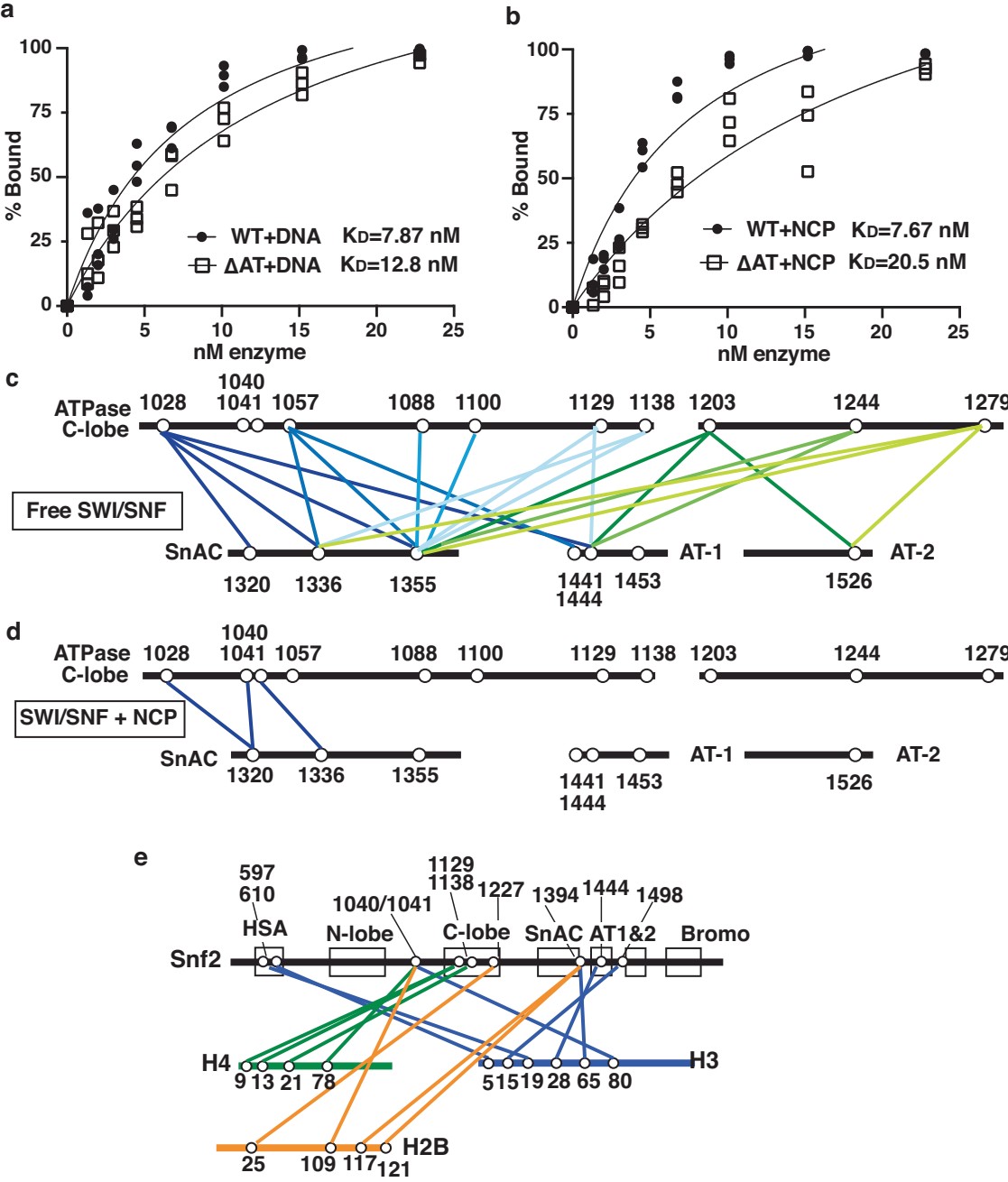

**Fig. 3 | The AT-hooks assist in SWI/SNF docking onto nucleosomes. a, b** The relative affinity ($K_D$) of WT and ΔAT SWI/SNF for (**a**) free DNA or (**b**) nucleosomes was measured by EMSA on a 4% native polyacrylamide gel (79:1 acrylamide:bisacrylamide). Reactions contained either 6.7 nM 235 bp 601 DNA or 29N59 nucleosomes and WT and ΔAT SWI/SNF was varied from 1.3 to 23 nM. **c–e** The Lys-Lys crosslinking patterns between (**c**) Snf2 and histones, (**d**) the C-lobe of the ATPase and SnAC domain and (**e**) between the SnAC domain and the AT-hooks are shown for nucleosome-bound SWI/SNF. The open circles indicate the positions of the crosslinked lysines, and the colored lines show the Lys-Lys crosslinked pairs. In (**a, b**) three replicates are performed and individual data points are shown. Source data are provided as a Source Data File.

inhibition (Fig. 4). The different phenotypes of the AT-hook and SnAC domain deletions versus *snf*2Δ and *snf*2-K798A suggest important context differences that may vary based on the extent to which SWI/SNF is inhibited. High efficiency of SWI/SNF remodeling may be required more for Gcn4-mediated gene activation and less when using ethanol or raffinose as an energy source.

## The AT-hook is required in mammalian SWI/SNF for stage-specific activation of enhancers

The AT-hook is a motif evolutionarily conserved in all eukaryotic versions of SWI/SNF which prompted us to examine the mammalian SWI/SNF complex where there is a single AT-hook by deleting exon 33 containing the AT-hook in both copies of *Brg*1 spanning residues 1401-1423 by CRISPR/Cas9 in mouse embryonic stem cells (mESCs). The genome-wide binding patterns of WT and two independent clones of the Brg1 AT-hook deletion mutant (ΔAT1 and ΔAT2) are mapped in two distinct pluripotent states referred to as naive and primed, representative of the pre-and post-implantation stages using Brg1 ChIP-seq. We observe Brg1 binds sites unique to either the naive or primed stages (Fig. 5a and Supplementary Fig. 4a). At those sites where Brg1 is bound in both the naive and primed stages, we observe a higher enrichment of Brg1 in the primed state (Fig. 5a). Brg1 is assembled primarily into

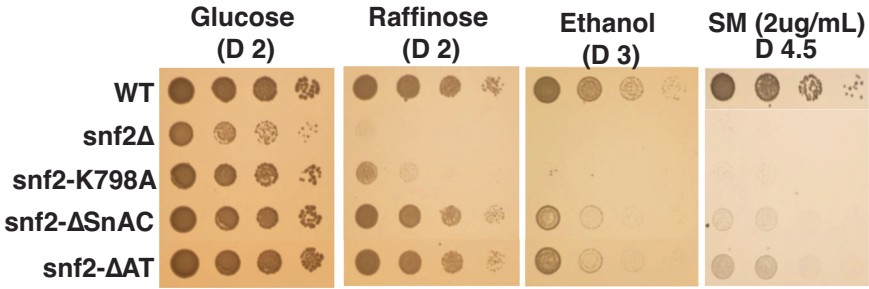

**Fig. 4 | The AT-hooks of Snf2 are needed in vivo for active yeast SWI/SNF.** Spot growth assays are shown for determining how deletion of the AT-hooks compare to deletion or other mutations in the Snf2 subunit known to impact the in vivo activity of SWI/SNF. Cells are grown in complete synthetic media plus sulfometuron methyl (SM) or other media with the following carbon sources: ethanol, glucose and raffinose.

esBAF (embryonic BAF) and GBAF or ncBAF (non-canonical BAF) complexes with very little PBAF present in mESCs, and the esBAF preferentially binds to cis-regulatory regions and GBAF to promoter regions[44]. Approximately 70-80% of the naive- and primed-specific Brg1 peaks are at intronic or intergenic regions and correspond to the esBAF complex and 13-18% of the peaks are associated with promoter regions (Supplementary Fig. 4b). Similar to ySWI/SNF, deletion of the AT-hook does not affect SWI/SNF complex integrity in either naive or primed cells as detected by Western blot and mass spectrometry respectively, although a modest reduction in the Arid1a and 1b subunits in the ΔAT complex in primed cells is observed (Supplementary Fig. 4c, e). The expression and transport of Brg1 into the nucleus is also not affected by loss of the AT-hook (Supplementary Fig. 4d).

Next, we focused on the intronic and intragenic regions to identify those that are active enhancers by mapping TF binding/DNA accessibility (ATAC-seq)[45] and the localization of acetylated lysine 27 (H3K27ac) and monomethylated lysine 4 (H3K4me1) of histone H3 by ChIP-seq. A total of 38,416 and 42,303 ATAC peaks are detected in naive and primed cells of which 18,814 peaks are in common between naive and primed (Supplementary Fig. 5a). The majority of state specific ATAC peaks reside in intronic and intergenic regions, approximately 14,000 peaks in naive or primed cells, and based on their co-localization with histone marks H3K27ac and H3K4me1 are in active enhancers (Fig. 5b and Supplementary Fig. 5b). Motif analysis reveals the accessible sites reflect binding of stage-specific TFs with the pluripotency TFs Oct4, Nanog, Klf family, and Sox2 being highly enriched at naive- and primed-specific sites and epiblast-specific TFs Zic2, Zic3, Otx2, and Glis3 being highly enriched at primed specific sites[46–49] (Supplementary Fig. 5c, d). There are two classes of pluripotency TF binding sites with some being unique to the naive stage and other sites are bound in both stages (Supplementary Fig. 5e).

There are more than 5000 stage-specific sites in either the naive or primed stages that depend on the AT-hook of Brg1 for TF binding as detected by ATAC-seq (Supplementary Fig. 5f). There are also about 7000 sites where TF binding is retained in either the naive or primed stage when the AT-hook is deleted (Supplementary Fig. 5g). Core pluripotency TFs like Oct4, Sox2, Nanog and Tcf bind to naive specific sites in an AT-hook dependent manner as observed by DNA footprint analysis (Fig. 5c). There are other sites where the pluripotency TFs bind that are occupied in both the naive and primed stages do not depend on the AT-hook of Brg1 for binding (Supplementary Fig. 5f). Zic3, an epiblast-specific TF, is dependent on the AT-hook for its binding, while Otx2, another epiblast-specific TF, is not dependent (Fig. 5e, f). A more thorough breakdown of the AT-hook dependency of naive- and primed-specific TF binding is shown in Supplementary Fig. 5f. While Brg1 had been shown previously to be required for recruitment of Oct4, Sox2 and Nanog, this is the first time Brg1 and its AT-hook is shown to promote binding of these TFs specifically in the naive state and to promote binding of a select set of epiblast-specific TFs in the primed state[19,50–52].

Beyond mediating TF binding, the connection of Brg1 and its AT-hook to enhancer activation is also examined for the AT-hook dependency of H3K27ac and H3K4me1. Monomethylation of lysine 4 of histone H3 at the naive- and primed-specific enhancers requires the AT-hook of Brg1; whereas acetylation of lysine 27 of histone H3 is not reduced by loss of the AT-hook (compare Figs. 6a, b to 6c, d). There is no tight connection between TF binding and H3K4me1 as H3K4me1 is lost at sites where TF factor binding is not affected (Fig. 6a, b compare upper and lower panels). Brg1 recruitment at these sites is not changed by loss of the AT-hook and these data demonstrate the catalytic activity of Brg1 is required for monomethylation of H3K4 and not for acetylation of H3K27 (Supplementary Fig. 6).

### Brg1 and its AT-hook are required to activate transcription in the naive and primed stages

To determine if these altered enhancers are indeed involved in transcription regulation, we map actively transcribing RNAP II with base-pair resolution using Precision Run-On Sequencing (PRO-seq)[53]. We observe there are different types of transcriptional changes that occur in the transition from the naive to primed stages. We observe many genes that are upregulated in this transition with some having increased promoter-proximal paused and elongating RNAPII that encoded for genes involved in development, differentiation, cell communication and signaling/transport (Fig. 7a). The second set of genes are up regulated by the release of paused RNAPII and encoded genes involved in various aspects of metabolism (Fig. 7b). Upregulation of both these sets of genes are consistent with the changes required for the transition to cell lineage priming. A third smaller set of genes are upregulated that only have increased paused RNAPII (Fig. 7c). There are other genes in the naive-to-primed transition that have only paused RNAPII or both paused and elongating RNAPII in the naive stage that are shut down in the primed stage (Supplementary Fig. 7c, d). This last set of genes encode for factors involved in oxidative phosphorylation (OX-PHOS) and pentose phosphate pathways (PPP) whose expression is repressed in the transition from naive to primed accompanying the shift to glycolysis[54,55], lipid metabolism[56–58], cell cycle control[59] and G protein-coupled receptor signaling[60,61] and have a pivotal part in exiting pluripotency.

When the AT-hook is deleted, there are genes that fail to be activated in the primed stage that encodes for some of the factors involved in cell development and differentiation (Fig. 7d and Supplementary Fig. 7a). These genes fall into two of the prior classes with some interfering with the formation of the promoter-proximal paused transcription only and others that increase both paused and elongating RNAPII (examples shown in Fig. 7f, g). These changes in nascent transcription reflect well the changes in enhancer activation that are observed earlier when the AT-hook mutant is deleted and points to Brg1 and its AT-hook regulating gene transcription through enhancer activation. There are also a second set of genes impacted by deletion of the AT-hook that fail to be shut down in the primed stage and are first

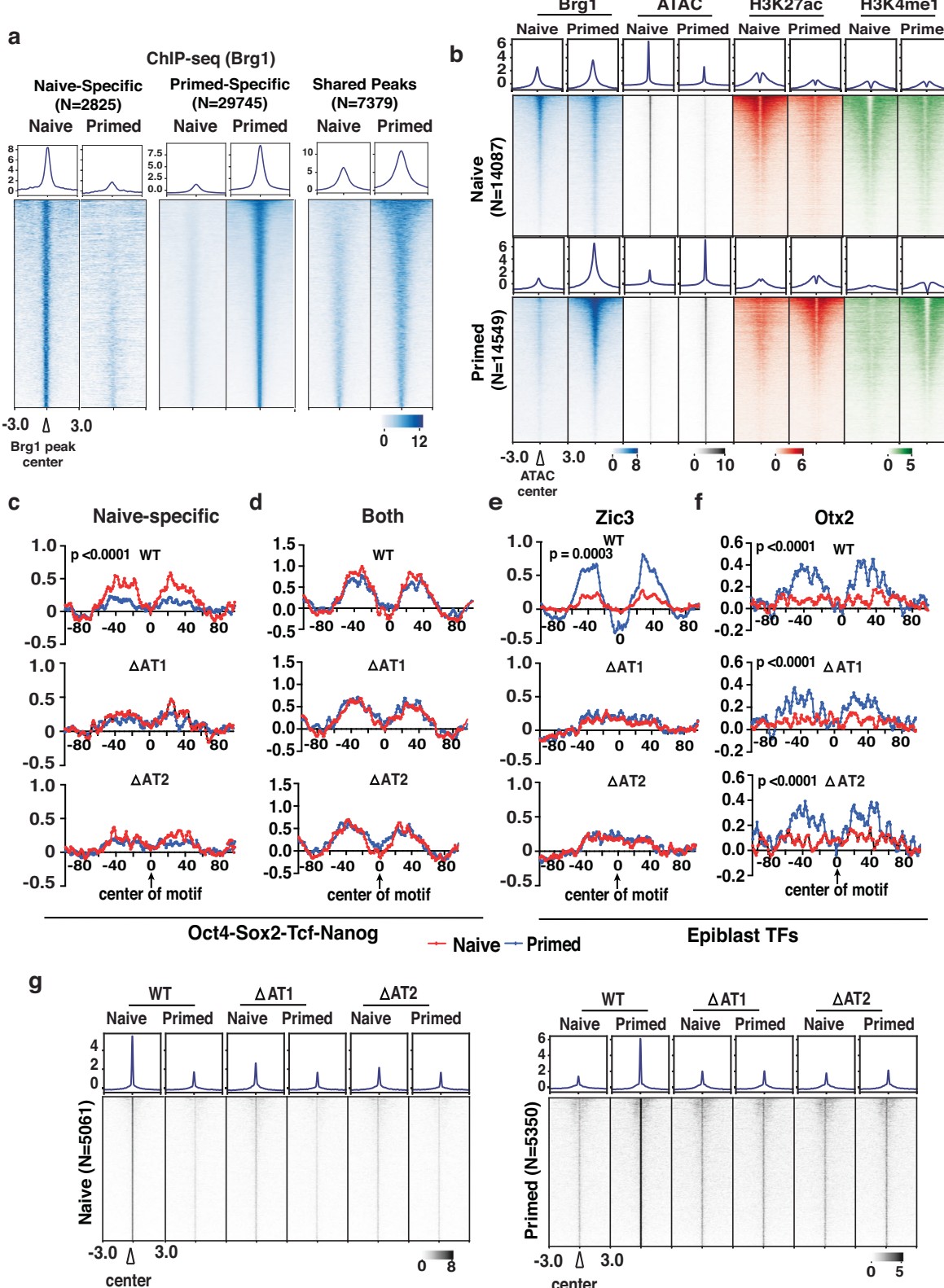

expressed in the naive stage and are involved in metabolism (Fig. 7e and Supplementary Fig. 7b). Both types of genes regulation are important for cell lineage priming and suggest that Brg1 has a role in gene repression as well as gene activation.

There are also transcriptional changes that occur in the naive state when the AT-hook is absent that fall into one of two categories. Many genes fail to be activated when the AT-hook is deleted and encode for

factors involved in cell communication, signaling, and some metabolic processes (Supplementary Fig. 7e, f). The other set of genes are those which are aberrantly upregulated in the naive stage (Supplementary Fig. 7g, h).

At the promoters of these genes, enrichment of trimethylated lysine 4 of histone H3 and DNA accessibility is not altered upon deletion of the AT-hook, consistent with gene dysregulation being due to

**Fig. 5 | The AT-hook of Brg1 is required for binding of stage-specific transcription factors to enhancers. a** Heatmaps show Brg1 localization that is unique to either the naive (left) or primed (middle) stage or are present in both stages, referred to as shared (right). **b** Brg1 localization (blue), DNA accessibility (ATAC-seq, grey), and active enhancer histone marks (H3K27ac [red], and H3K4me1 [green]) are shown for naive- (top) and primed- (bottom) specific intronic-intergenic sites. ChIP-seq signals are sorted based on WT Brg1 in each condition. **c**–**f** DNA footprinting shows loss of (**c**) naive-specific pluripotency TFs Oct4, Sox2, Tcf and Nanog and (**e**) primed/EpiSC specific TF Zic3 binding in the AT-hook

deletion mutants. DNA footprinting of (**d**) pluripotency TFs Oct4, Sox2, Tcf and Nanog and (**f**) EpiSC specific Otx2 binding sites are shown that remain unaltered in the ΔAT mutants. Both copies of *Brg*1 had its single AT-hook removed by deleting residues 1401–1423. Statistical significance was calculated by Wilcoxon test and Paired t-test for OTSN (naive specific and shared), Zic3 and Otx2 motifs respectively. **g** The sites where DNA accessibility related to TF binding is shown for sites that depend on the AT-hook in either the naive (left) or primed (right) stages. ATAC signals are sorted based on WT. Source data are provided as a Source Data File.

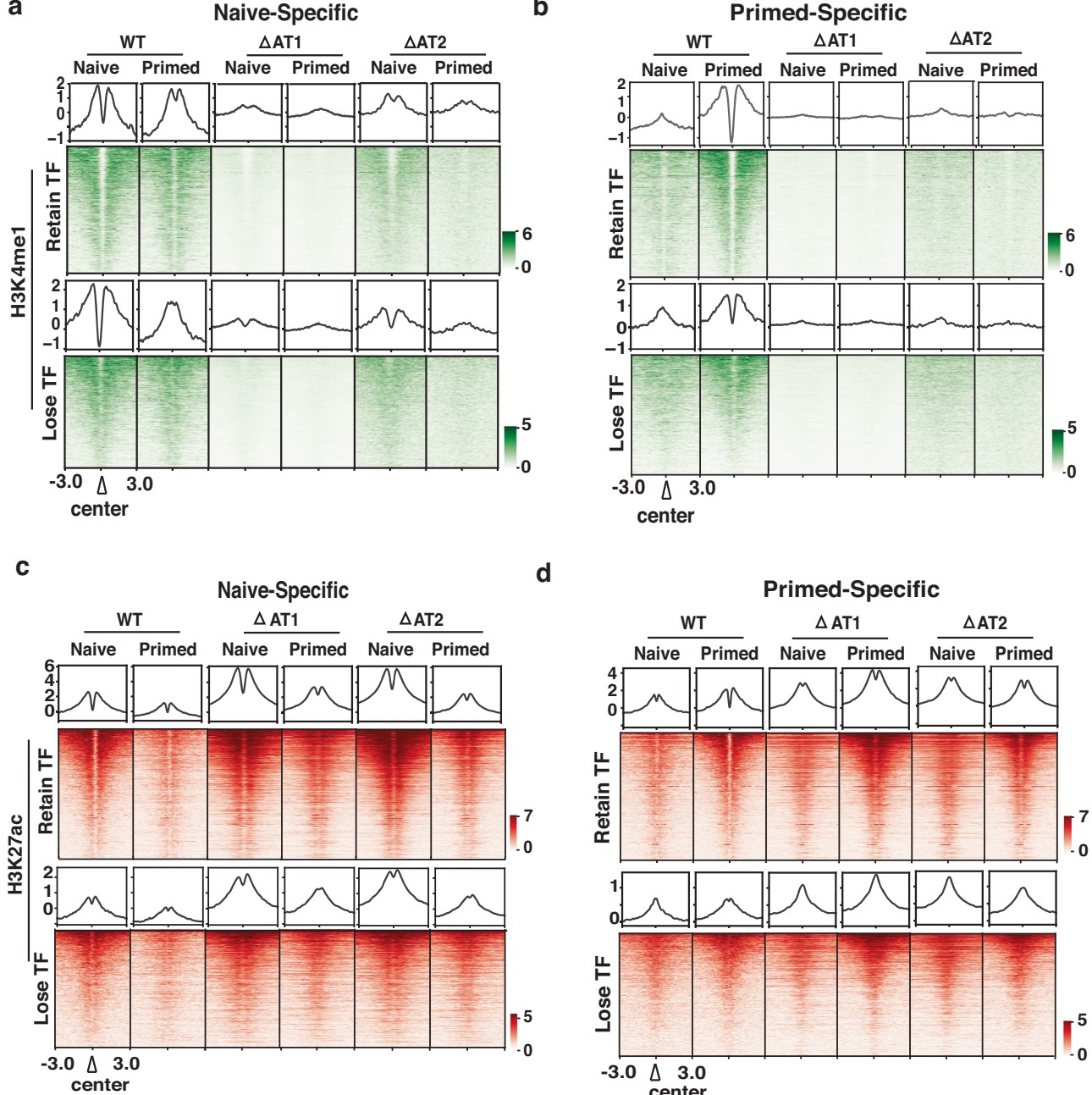

**Fig. 6 | Monomethylation of H3K4 at stage-specific enhancers is dependent on the AT-hook of Brg1. a**, **b** H3K4me1 localization at (**a**) naive- and (**b**) primed-specific enhancers are shown. The regions are divided into those where the DNA accessibility (ATAC) is either AT-hook dependent (top panel) or independent (bottom panel). ChIP signals are sorted based on WT in each group and on naive

and primed WT in each group. **c**, **d** H3K27ac localization at (**c**) naive- and (**d**) primed-specific enhancers is shown. The regions are divided into those where the DNA accessibility (ATAC) is either AT-hook dependent (top) or independent (bottom) as in (**a**, **b**). ChIP signals are sorted based respectively on naive and primed WT in each group.

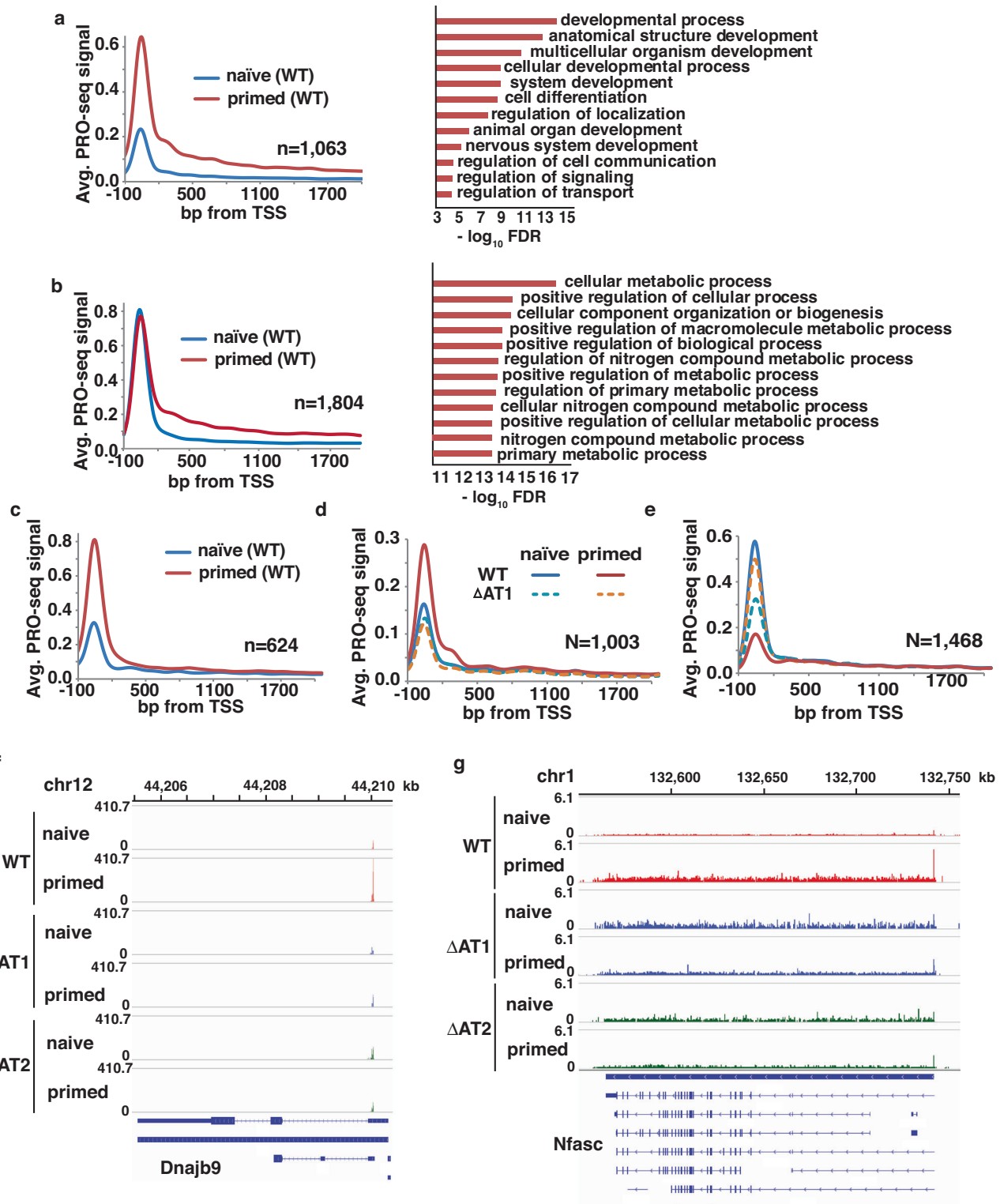

**Fig. 7 | Loss of the AT-hook of Brg1 causes transcription dysregulation in both the naive and primed states. a–c** PRO-seq metagene-analysis is shown for genes upregulated in the primed stage for WT cells (upstream TSS −100 bp to +300 bp downstream of TSS). The genes in (**a**) have increased formation of initiation transcription complexes and genes in (**b**) have the release of promoter-proximal paused RNAPII shown on the left with the corresponding gene ontology analysis shown on the right. Genes in (**c**) have increased formation of promoter-proximal paused RNAPII complexes as in (**a**), except there is negligible elongation in (**c**). **d, e** PRO-seq metagene analysis is shown for WT and ΔAT1 for genes that are dysregulated in the primed stage and either (**d**) fail to be activated when the AT-hook is deleted in the primed stage or (**e**) fail to be shut down in the transition from naive to primed. **f, g** The PRO-seq pattern for (**f**) Dnajb9 and (**g**) Nfasc genes are shown that represent two classes of dysregulated genes when the AT-hook is deleted that are important in cell fate determination.

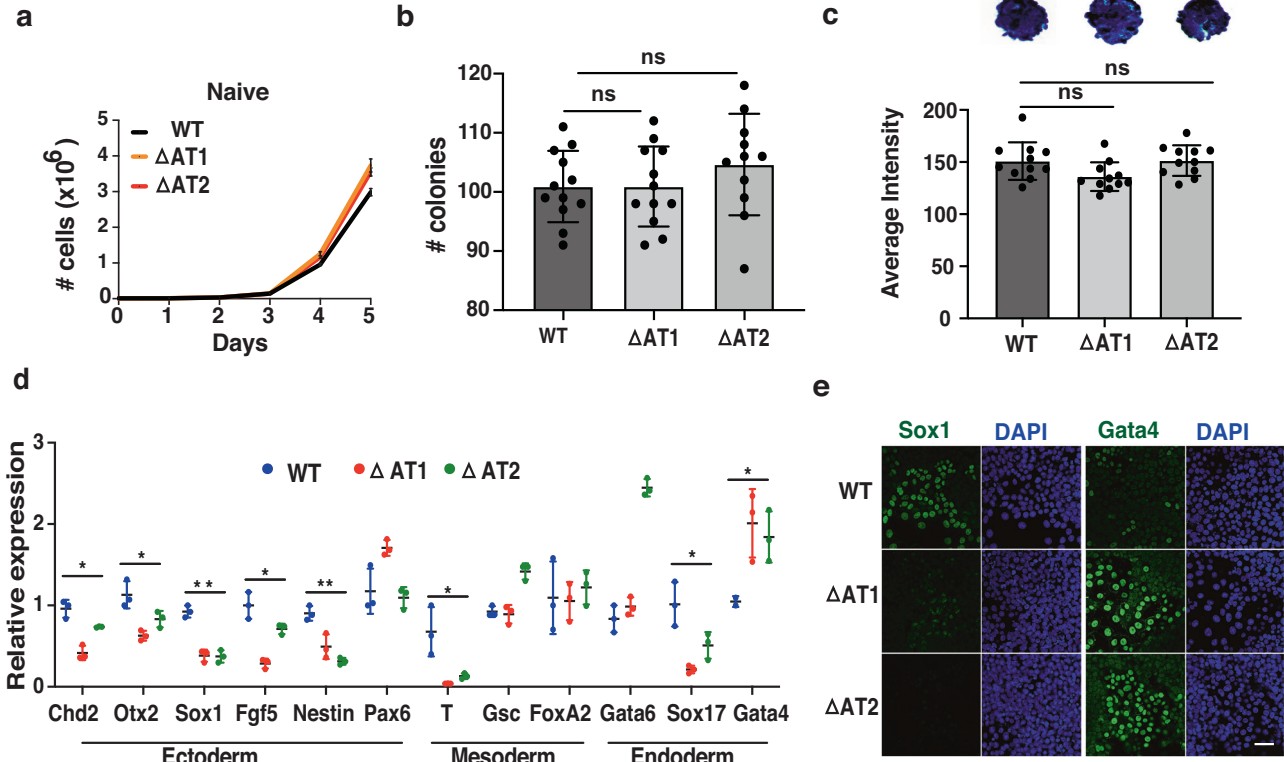

**Fig. 8 | The AT-hook domain of Brg1 is required for cell lineage priming.**
**a** Growth curves of WT and AT-hook deletion mutant clones (ΔAT1 and ΔAT2) cultured in naive condition are shown. **b**, **c** Bar graphs showing **b** the number of colonies formed in a self-renewal assay and **c** the average signal intensity of the colonies in alkaline phosphatase assay of WT and ΔAT mESCs in naive condition. Results are presented as means ± sd (n = 12 for panel (**b**) and n = 11 for panel (**c**)). Unpaired t-test and one way ANOVA was done for panel b and c respectively. **d** Bar graph shows the expression of lineage-specific markers in WT and ΔAT mutants

after culturing for seven days in the absence of LIF/2i. Gene expression analysis was done by quantitative reverse transcription PCR (qRT-PCR) and the values were normalized with GAPDH. Results are presented as means ± sd (n = 3); *p < 0.05; **p < 0.001; (unpaired student's t-test). **e** Immunofluorescence images showing expression of Sox1 (ectoderm lineage marker) and Gata4 (endoderm lineage marker) in WT and ΔAT mutants after seven days of LIF/2i withdrawal. Scale bar, 20 microns. Source data are provided as a Source Data File.

altered enhancer activity rather than restructuring of promoters (Supplementary Fig. 8a–f). The AT-hook dependent enhancers are in the correct proximity (250 kb-1 Mb) to regulate these genes in naive and primed cells (Supplementary Fig. 8g). In summary, we find the AT-hook of Brg1 is vital for Brg1's role in activating stage-specific enhancers which regulates expression of genes important in differentiation and cell development.

**Loss of the AT-hook from Brg1 disrupts cell lineage priming**
Although many transcriptional changes occur in the naive state when the AT-hook is deleted, these do not affect stem cell proliferation, clonal expansion, alkaline phosphatase staining or colony morphology (Fig. 8a–c and Supplementary Fig. 9a–c). The expression and nuclear localization of the core pluripotency TFs Oct4, Nanog and Sox2 are also not altered by loss of the AT-hook (Supplementary Fig. 9c). These results highlight the differences between the complete loss of Brg1 versus loss of the AT-hook since the absence of Brg1 or chemical inhibition of its catalytic activity blocks expression of the core pluripotency TFs and stem cell maintenance and proliferation[62,63].

Brg1 and its AT-hook are however important for cell lineage priming as shown by measuring the expression of three distinct classes of lineage-specific markers by qRT-PCR after culturing cells in the absence of LIF and two inhibitors for 7 days. Ectodermal markers Sox1 and Nestin, mesoderm marker Tbxt and the endoderm marker Sox17 are all down regulated in both ΔAT mutant clones compared to WT cells, consistent with defects in cell lineage priming (Fig. 8d). The endoderm-specific gene Gata4 is aberrantly highly expressed in both

clones of the ΔAT mutants compared to wild type (Fig. 8d). The intracellular levels of Sox1 and Gata4 in the two ΔAT mutant clones are respectively lower and higher in wild type cells as shown by immunofluorescence, thus suggesting the ΔAT mutant has a bias toward differentiating into endoderm compared to wild type (Fig. 8e). These results further confirm the importance of the AT-hook in cell lineage priming consistent with our profiling of enhancer activation and gene regulation in the primed state.

## Discussion
We observe the AT-hook regulates the nucleosome remodeling activity independent of facilitating SWI/SNF recruitment to nucleosomes, which is different than originally proposed for chromatin-modifying proteins containing an AT-hook[3,4]. The intrinsic DNA-dependent ATPase activity of SWI/SNF depends on the AT-hook as seen using free DNA and yeast SWI/SNF. The reduction in ATPase activity is directly coupled to a reduction in nucleosome mobilization in contrast to the uncoupling previously observed for the SnAC domain[64]. The AT-hook has a modest secondary role in enhancing SWI/SNF affinity for nucleosomes, which could potentially be due to its interaction with the N-terminal tail of histone H3. However, given the proximity of the bromodomain to the AT-hook, the bromodomain because of its ability to bind H3K14ac could be a bridge between the AT-hook and histone H3 tail[65].

The AT-hook is also shown to be important for the in vivo activity of SWI/SNF in yeast and mouse embryonic stem cells and has provided novel insights into SWI/SNF not observed previously by either deleting its catalytic subunit or completely inhibiting its ATPase/remodeling

activity. Our studies show attenuation of SWI/SNF activity can have outcomes that are distinct from complete loss or shutdown of SWI/SNF and has helped uncover for the first time Brg1 being required for early cell lineage priming. Whereas deletion of Brg1 blocks stem cell proliferation/self-renewal and expression and recruitment of the core pluripotency transcription factors for positive feedback regulation; deletion of the AT-hook does not[19,50,62]. The pluripotency circuitry shared in naive and primed states where the pluripotency TFs are bound at the same sites in both states are not perturbed by loss of the AT-hook. The AT-hook is instead required for stage-specific binding of pluripotency TFs to naive-specific sites and a subset of epiblast TFs to primed-specific sites. We don't know yet the difference between the two sets of pluripotency TFs binding sites that cause them to have different SWI/SNF requirements. Similarly, we find the co-dependency between Brg1 and H3K27ac is not dependent on the remodeling efficiency of Brg1 and therefore on the AT-hook[22,23,66]. Surprisingly, we observe H3K4me1 at stage-specific enhancers is dependent on efficient SWI/SNF remodeling and is not required for H3K27ac at these sites. They are due to catalytic differences since there is no change in Brg1 localization at these sites when the AT-hook is deleted and the yeast SWI/SNF biochemical data clearly shows the AT-hook regulates the catalytic activity.

We also observed in yeast that attenuation of SWI/SNF activity by the AT-hook reveals in vivo aspects of SWI/SNF not observed by deletion of the catalytic subunit or complete shutdown of its enzymatic activity. We observed the phenotypes well known for SWI/SNF dependency to be variable when the AT-hook is deleted, thus making it possible to subdivide these based on how strictly they depend on the remodeling efficiency of SWI/SNF. For example, switching from glucose to raffinose does not depend on the efficiency of SWI/SNF to mobilize nucleosomes, whereas Gcn4 activation of amino acid biosynthetic genes is dependent.

We show for the first time by extensive biochemical, genetic, and genomic analyses the importance of attenuating SWI/SNF remodeling and the role of the AT-hook in this process. These studies highlight the need to not only study SWI/SNF in vivo by subunit deletion or chemical inhibitors to completely shut down its enzymatic activity, but to also keep in mind the power of attenuation of SWI/SNF for the in vivo regulation of gene expression and chromatin structure.

## Methods

### Yeast strains and SWI/SNF preparation
Mutant SWI/SNF strains were generated using a yIpLac128 plasmid containing Snf2 along with C terminal double Flag epitope and LEU2 marker. Domain deletions were generated by gap repair where 2 PCR products flanking the domain of interest (either both AT hook domains or only the N terminal AT hook domain) were made using yIpLac128-SNF2-2FLAG-LEU2 plasmid as a template and were co-transformed into a Δsnf2 strain. Primer pairs used for generating the PCR products to delete both N and C terminal AT hooks (ΔAT) are summarized in Supplementary Table 2 and the complete list of yeast strains are shown in Supplementary Table 1. After the co-transformation of both primer pairs, yeast was grown in a Synthetic complete (-Leu) plate and replica plated onto a YPD-Kanamycin plate. Colonies that grew on the Sc-Leu plate but not on the YPD-Kan plate were then streaked onto a fresh Sc-Leu plate and sequenced for confirmation of the domain deletion and correct integration of the mutant snf2 sequence into the genome at the N- and C-terminus of the point of insertion. Clones which were sequenced, and integration positive were then grown separately in 100 ml YPD media and expression of the mutant Snf2 confirmed by Western blot with Anti-Flag antibody. SWI/SNF was purified as described previously in ref. 67.

### Nucleosome reconstitution
Homogenous mononucleosomes were reconstituted with 10 ug of PCR-generated DNA using p159 plasmid containing a 601 nucleosome positioning sequence (NPS) as template and 29 bp and 59 bp flanking either side of the 601 sequence (29N59), 10 μg of recombinant *Xenopus laevis* histone octamer and 100 fmol of $^{32}P$ labeled 29N59 DNA at 37 °C by a rapid salt dilution method[68]. The *Xenopus laevis* histone octamer was made as described[69]. Heterogenous nucleosomes were reconstituted with 100 fmol of $^{32}P$ labeled 29N59 DNA, 10 μg of histone octamer, and 9 μg of salmon sperm DNA by the same method. The radiolabeled DNA was generated by PCR using an oligonucleotide labeled using Optikinase (USB) and γ $^{32}P$ ATP (6000 Ci/mol).

### Electrophoretic mobility shift assays (EMSA)
A titration was done with purified SWI/SNF (WT or mutant) pre-bound to homogenous/heterogenous nucleosomes (20 nM) or DNA (10 nM) at 30 °C for 15 mins in the presence or absence of 3.2 nM Gal4-VP16 to determine the amount of SWI/SNF at which >90% of the nucleosome (or DNA) was fully bound as seen by gel-shift in a 4% native polyacrylamide gel (acrylamide:bis-acrylamide ratio 37.5:1) run at 200 V at 0.5x Tris-Borate-EDTA. Using these conditions, remodeling assays were performed where after prebinding for 15 mins at 30 °C, ATP was added to a final desired concentration and then the reaction stopped at defined time points by the addition of salmon sperm DNA and EDTA (final concentration 0.5 μg/ μl and 40 mM respectively in reaction). The remodeled products were analyzed on 5% native polyacrylamide gels (acrylamide:bis-acrylamide ratio 60:1) at 100 V in 0.5x Tris-Borate-EDTA.

### ATPase assays
ATPase kinetic assays were performed similar to remodeling assays with γ $^{32}P$ ATP (0.02 μCi) mixed with cold ATP with homogenous Xenopus mononucleosomes or with 100 bp pUC18 plasmid DNA. Reactions were stopped using a mixture of SDS and EDTA (final concentration 2.5% and 50 mM in reaction respectively). One μl of stopped reactions were spotted onto JD Baker PEI-Cellulose TLC plates and run in 0.8 M glacial acetic acid, 0.8 M lithium chloride. Plates were dried and imaged by autoradiography.

### Growth assays
WT and mutant SWI/SNF were grown overnight in 5 ml YPD cultures and the OD600 was adjusted to 1 by the addition of fresh YPD. 100 ul of each culture was taken out and 5-10-fold serial dilutions were made. Ten ul of the original culture (at OD600 = 1) and each of the serial dilutions were then aliquoted onto YPD, glucose, raffinose, ethanol, and SM plates and dried. Plates were then turned over and left in 30 °C for 2–3 days until appearance of yeast colonies.

### Chemical crosslinking and mass spectrometry analysis[70]
SWI/SNF with Snf2 tagged at its C-terminus with two copies of the FLAG peptide was purified in a two-step process. SWI/SNF was purified by M2 immunoaffinity chromatography and the FLAG peptide removed by cation exchange chromatography using SP-Sepharose[70]. Crosslinking reactions contained 35 nM FLAG tagged SWI/SNF, 35 nM 29-N-59 nucleosomes and 200 nM Gal4-VP16 in a final volume of 520 μl and BS3 (bis(sulfosuccinimidyl)suberate), an amine homobifunctional crosslinker from Thermo Scientific) was added to a final concentration of 115 μM. After crosslinking at room temperature for 2 h, the reaction was quenched by adding 20 μL of 1 M ammonium bicarbonate, precipitated by addition of trichloroacetic acid (TCA) to 16.7%. Proteins were resuspended in 100 μL 8 M urea in 2 M ammonium bicarbonate prior to reduction with 10 mM *tris*(2-carboxyethyl)phosphine (TCEP) and alkylation with 20 mM iodoacetamide. The urea concentration was reduced to 1 M by addition of 100 mM ammonium bicarbonate, and trypsin was added at a 20:1 (w/w) ratio for an overnight incubation at 37 °C. The resulting peptides were desalted on a C18 column and then fractionated by microcapillary strong cation exchange chromatography (200 μm inner diameter [I.D.] × 20 cm). Peptides were loaded

onto the column and washed with buffer A (20% acetonitrile [ACN], 0.1% formic acid [FA]). Peptides were eluted in four steps by increasing the percentage of buffer B (20% ACN, 1.0 M ammonium formate, pH 3.5): 30% B, 50% B, 70% B, and 100% B. A final elution with 10% ACN, 500 mM ammonium acetate was also used. Each fraction was dried and analyzed directly by mass spectrometry. The integrity of the SWI/SNF-nucleosome complex was tested after crosslinking by SDS-PAGE to ensure BS3 did not disrupt SWI/SNF binding.

RAW files were converted to mzXML by RawConverter software. We used the Comet search engine and the Trans-Proteomic Pipline (TPP, http://tools.proteomecenter.org/ wiki/index.php?title=Software:TPP) were used for identification of unmodified and BS3 mono-modified (monolinks) peptides. Two crosslink database search algorithms were used for crosslinked peptide searches: pLink20and in-house designed Nexus with default settings. The database used for crosslink identification contained the wild type and modified protein sequences of the yeast SWI/SNF subunits and their reverse decoy sequences. A 5% FDR cutoff was used for both pLink and Nexus searches. The results were combined, and each spectrum was evaluated for the quality of the match to each peptide using the COMET/Loriket Spectrum Viewer (TPP). Crosslinked peptides were considered confidently if the majority of the observed ions are accounted for and at least 4 consecutive b or y ions were observed for each peptide. Finally, interaction plots are made using Circos online tool[71].

## Mouse embryonic stem cell culture

Mouse embryonic stem cells (E14Tg2a, ATCC) were maintained on 1% gelatin-coated plates in the ESGRO complete plus clonal grade medium (Millipore), as previously described[72,73]. Embryonic stem cells (ESCs) were cultured on gelatin-coated plates in DMEM (Invitrogen) supplemented with 15% FBS, 1X-Gultamax (Gibco), Na-Pyruvate (Gibco), 10 mM 2-mercaptoethanol, 0.1 mM nonessential amino acids (Gibco), 1U/ml of ESGRO mLIF (Millipore), and 2i inhibitors (MEK inhibitor PD0325901, Gsk3b inhibitor (CHIR99021 – Stem Cell technology) in naive (or +2i) condition. In epiblast stem cells (EpiSCs) or primed condition, cells were cultured in chemically defined medium (IMDM and F12-Invitrogen) supplemented with 2%-BSA (Sigma), Insulin (Roche), Transferrin (Roche), CD-lipid concentrate (Gibco), FGF2 (R&D), and Activin-A (R&D) (Ref). For spontaneous differentiation (SD), cells were cultured in gelatin-coated plate in previously described DMEM-FBS media without LIF and 2i-inhibitors. Cell lines were continuously monitored under a microscope and confirmed to be free of mycoplasma contamination by using a MycoAlert mycoplasma detection kit (Lonza) and DAPI staining.

## CRISPR mediated deletion of the AT-hook of *BRG1* and insertion of the HA-tag

Guide RNA used for CRISPR-Cas9 editing were designed using the CRISPR Design Tool (http://crispr.mit.edu) to minimize off-target effects. G-blocks containing the guide RNA directed to the AT-hook domain of Smarca4/Brg1 (exon 30, targeting amino acids 1401 to 1423) came from IDT and were PCR amplified, cloned into two Cas9 containing plasmids (pX330; Addgene#158973) using Zero Blunt TOPO Cloning Kit (Invitrogen) and sequence verified. After 72 h of transfection in mESCs using electroporation, positive colonies were selected based on puromycin (1 mg/ml) selection. Plasmids were separately tested in trial transfection in E14 cells to determine the efficiency of guide RNA cleavage by isolating genomic DNA (QuickExtract DNA Extraction from Epicentre), the region of interest being amplified by PCR and editing identified using EnGen Mutation Detection Kit (NEB). All guide RNA used in HA-tag insertion or AT-hook deletion had a cleavage efficiency of >60%. ESCs were seeded at low density to allow for the selection of individual colonies. Colonies were individually expanded and split for future culture or genomic DNA isolation. Genomic DNA (100 ng) from these colonies was used to confirm the desired targeted deletion by PCR. The donor DNA for HA-tag insertion contained a BamHI cut site that was used for screening for positive clones. We obtained several clones with homozygous and heterozygous insertion of the HA-tag. We used a single BfuAI cut site in the AT-hook region to screen for positive AT-hook deletion clones and from 129 clones obtained 10 homozygous knockouts of the AT-hook in E14 cells.

## Cell proliferation, self-renewal assay, and alkaline phosphatase staining

Growth assay was done with $1 \times 10^4$ cells seeded in gelatin-coated six-wells plate and monitored for the next 5 days in the respective culture condition (i.e. naive and primed). Cell counting was done every other day using hemocytometer and viability was checked using trypan blue at the time of counting. Self-renewal and alkaline-phosphatase staining assays were done with 200 cells seeded in gelatin-coated 12-wells plate and maintained in naive media condition. Colonies were counted and stained with alkaline phosphatase (ALP) after 5 days. ALP staining was done following the manufacturer's protocol (SBI-AP100B-1). Media was changed every other day.

## Immunofluorescence staining

Cells were seeded in poly-ornithine (EMD-Millipore) and laminin (Sigma) coated 8-chambered slide (ibidi; cat#80806) and maintained in naive and differentiation media independently. Next, cells were fixed with 4%-paraformaldehyde for 15 min at room temperature (RT) and washed with 1XPBS and permeabilized with 0.5% Triton for 5 min at RT. Blocking was done for 30 min at RT with 5% bovine serum albumin (Sigma). Primary antibodies were used with recommended dilutions and incubated for overnight at 4 °C. After primary antibody incubation, cells were washed with 1XPBS (Phosphate buffer solution) and 1XPBST (PBS + 0.1%Tween20) and then follow secondary antibody incubation. For immunostaining, antibodies including anti-Oct4 (abcam; ab107156; dilution 1:250), anti-Sox2 (abcam; ab107156; dilution 1:500), anti-Nanog (abcam; ab107156; dilution 1:200), anti-Brg1/Smarca4 (abcam; ab110641; dilution 1:100), anti-Sox1 (CST; 4194 S; dilution 1:200), anti-Gata4 (scbt; sc-25310; dilution 1:50) were used. Images were captured using a LSM-880 confocal microscope (Zeiss) and processed with Zen-blue software.

## RNA isolation and quantitative real-time PCR (qRT-PCR)

Total RNA was isolated using Trizol (Invitrogen), following the manufacturer's protocol. One mg of RNA was used to prepare cDNA and were synthesized using the iScript kit (Bio-rad) according to the manufacturer's protocol. For each biological replicate, quantitative PCR reactions were performed in technical triplicates using the iTaq Universal Sybr Green Supermix (Bio-Rad) on the Bio-Rad CFX-96 Real-Time PCR System, and the data normalized to *Gapdh*. Primers used in this study are summarized in Supplementary Table 2.

## Immunopurification of wild type and ΔAT mutant Smarca4/Brg1 complexes

Purification of HA-tag protein was performed following a previously described protocol ([16]). Briefly, cells (~ 2-3 × 10^8) were cultured in gelatin-coated plates and maintained in naive and primed media independently. The packed cell volume (PCV) was estimated after cell harvesting and gently resuspended in buffer with 10 mM HEPES-KOH pH 7.9, 10 mM KCl, 1.5 mM MgCl$_2$ 1 mM DTT, 1 mM PMSF, 1uM pepstatin, 10uM chymostatin. Next, the cell suspension was transferred to a Dounce homogenizer fitted with a B-type pestle, and the cells lysed with 20 strokes followed by centrifugation for 5 min at 900 x *g*. After removing the supernatant, the nuclei pellet was resuspended in buffer containing 0.2 mM EDTA, 20% Glycerol, 20 mM HEPES-KOH pH 7.9, 420 mM NaCl, 1.5 mM MgCl$_2$, 1 mM DTT, 1 mM PMSF, 1uM pepstatin, 10uM chymostatin, protease inhibitor cocktail and the pellet lysed by

gentle Dounce homogenization (i.e., 10–20 strokes with type-B pestle). The tube was mounted on a vortex mixer and agitated very gently for 30 minutes to 1 hr at 4 °C degree and centrifuged for 15 minutes at 20,000 x *g* and the supernatant collected. Nuclear lysates were diluted with two-thirds of original volume of 20 mM HEPES, pH 7.9, and 0.3% NP-40 to adjust to an appropriate NaCl concentration. Anti-HA agarose beads were used following the manufacturer's protocol[74] and the nuclear extract was incubated with HA-beads overnight at 4 °C with gentle end-over-end mixing or on a rocking platform. Next, beads were washed three times with wash buffer (50 mM Tris pH 8.0, 150 mM NaCl, 1 mM EDTA, 10% Glycerol, and 0.5% Triton X-100) and one bed volume of HA-peptide was added to the beads and incubated at 4 °C for 4 h prior for elution of the complex. Untagged cells were lysed using 1X lysis buffer (CST) following manufacturer's protocol to prepare the whole cell extract. The efficiency of the immunoprecipitation was checked by Western blotting following standard protocols with the following primary antibodies anti-Brg1/Smarca4 (abcam; ab110641; dilution 1:1000), anti-HA (Invitrogen; Cat #26183; dilution 1:1000), anti-Tubulin (ThermoFisher; A11126; dilution 1:1000), and anti-Gapdh (CST; 2118; dilution 1:2000). The immunoblot were visualized using Super Signal Pico chemiluminescent reagent.

## PRO-seq

PRO-seq was performed as previously described with minor modifications[53]. Briefly, nuclei were isolated using Dounce homogenizer (1 million cells per mL) and nuclear run-on was performed with all four biotin-NTPs. RNAs were extracted by Trizol LS (Ambion) and fragmentated by base hydrolysis. From the fragmentated RNAs, biotin RNAs were enriched by streptavidin beads. The biotin RNAs were enriched twice more in each 3′ and 5′ RNA adaptor ligation processes. Reverse transcription, PCR amplification, and library size selection were performed to obtain 140 - 350 bp, 5 ng/uL libraries. These libraries were submitted for sequencing (75 bp, single-endreads) on an Illumina NextSeq 500.

## ChIP-seq

Chromatin immunoprecipitation (ChIP) was performed following previously described high-throughput ChIP protocol with some modifications[75]. Cells were crosslinked in 1% formaldehyde for 10 min at room temperature, before quenching with 125 mM glycine for 5 min. After 2 washes with ice-cold PBS, cells were incubated in lysis buffer (5 mM PIPES pH 8.0, 85 mM KCl, 0.5 % NP-40 supplemented with protease inhibitor) for 10 min, and nuclei were collected after centrifugation. The nuclear pellet was re-suspended in shearing buffer (12 mM Tris-HCl pH 7.5, 6 mM EDTA pH 8.0, 0.5 % SDS supplemented with protease inhibitor) and chromatin was fragmented using ME220 focused ultra-sonicator (Covaris) to obtain DNA fragments ranging 200-600 bp. The chromatin lysate was collected after centrifugation and incubated overnight at 4 °C with Brg1 and respective histone antibodies conjugated with Dynabeads Protein G (Invitrogen). Next day, antibody-bound DNA were collected using Dynamag, washed extensively as described in the protocol, treated with RNase and Proteinase K, and reverse crosslinked overnight followed by DNA extraction using Ampure X beads (Beckman Coulter). Purified ChIP DNA was used for library construction using NEB Ultra II DNA library prep kit (New England Biolabs) and submitted for sequencing (75 bp paired-end and 50 bp single reads) on an Illumina HiSeq3000.

## ATAC-seq

ATAC-seq was performed as previously described[76]. Briefly, 50,000 cells were washed with cold PBS, collected by centrifugation then resuspended in resuspension buffer (10 mM Tris-HCl, pH 7.4, 10 mM NaCl, 3 mM MgCl2). After collection, cells were lysed in lysis buffer (10 mM Tris-HCl, pH 7.4, 10 mM NaCl, 3 mM MgCl2, 0.1% NP-40) and collected before incubating in a transposition mix containing Tn5 transposase (Illumina). Purified DNA was then ligated with adapters, amplified and size selected for sequencing. Library DNA was sequenced with paired-end 50 bp reads.

## FASP Methods – Orbitrap Exploris DIA

Protein samples were reduced, alkylated, and digested using filter-aided sample preparation with sequencing grade-modified porcine trypsin (Promega). Tryptic peptides were then separated by reverse phase XSelect CSH C18 2.5 um resin (Waters) on an in-line 150 × 0.075 mm column using an UltiMate 3000 RSLCnano system (Thermo). Peptides were eluted using a 60 min gradient from 98:2 to 65:35 buffer A:B ratio (Buffer A = 0.1% formic acid, 0.5% acetonitrile; Buffer B = 0.1% formic acid, 99.9% acetonitrile). Eluted peptides were ionized by electrospray (2.2 kV) followed by mass spectrometric analysis on an Orbitrap Exploris 480 mass spectrometer (Thermo). To assemble a chromatogram library, six gas-phase fractions were acquired on the Orbitrap Exploris with 4 m/z DIA spectra (4 m/z precursor isolation windows at 30,000 resolution, normalized AGC target 100%, maximum inject time 66 ms) using a staggered window pattern from narrow mass ranges using optimized window placements. Precursor spectra were acquired after each DIA duty cycle, spanning the m/z range of the gas-phase fraction (i.e., 496–602 m/z, 60,000 resolution, normalized AGC target 100%, maximum injection time 50 ms). For wide-window acquisitions, the Orbitrap Exploris was configured to acquire a precursor scan (385-1015 m/z, 60,000 resolution, normalized AGC target 100%, maximum injection time 50 ms) followed by 50 × 12 m/z DIA spectra (12 m/z precursor isolation windows at 15,000 resolution, normalized AGC target 100%, maximum injection time 33 ms) using a staggered window pattern with optimized window placements. Precursor spectra were acquired after each DIA duty cycle.

## Data-analysis

**PRO-seq analysis.** Adaptor trimming and low-quality reads were removed by Cutadapt[77]. The filtered reads were aligned on mm10 genome by bowtie2[78] with "−very-sensitive" option to discard reads mapped to more than one region. The mapped reads were compressed as binary form using samtools[79] and rRNAs were removed from those reads by bedtools intersect[80]. The 3′ end of the filtered reads were captured and in strand-specific manner using bedtools genomecov. These files were used to calculate raw readcount of annotated genes by bedtools map. Similarity between biological replicates and two AT hook deletion clones, and difference between samples were confirmed by pearson correlation with clustering and PCA analysis using R package DESeq2[81]. After the validation of replicates, all replicates were merged and normalized to meet final 30 million reads. The normalized reads were used to obtain differential genes through log2 fold-change calculation, create MA plots using R package ggplot2[82] and perform meta-plot analysis using deepTools2[83] with gaussian smoothing by R package Smoother[84]. Statistically significant gene ontology (GO) terms for each differential genes were annotated by GO enrichment analysis (Adaptor trimming and low-quality reads were removed by Cutadapt[77]. The filtered reads were aligned on mm10 genome by bowtie2[78] with "−very-sensitive" option to discard reads mapped to more than one region. The mapped reads were compressed as binary form using samtools[79] and rRNAs were removed from those reads by bedtools intersect[80]. The 3′ end of the filtered reads were captured and in strand-specific manner using bedtools genomecov. These files were used to calculate raw readcount of annotated genes by bedtools map. Similarity between biological replicates and two AT hook deletion clones, and difference between samples were confirmed by pearson correlation with clustering and PCA analysis using R package DESeq2[81]. After the validation of replicates, all replicates were merged and normalized to meet final 30 million reads. The normalized reads were used to obtain differential genes through log2 fold-change calculation, create MA plots using R package ggplot2 [82]and perform meta-plot analysis

using deepTools2[83] with gaussian smoothing by R package Smoother[84]. Statistically significant gene ontology (GO) terms for each differential genes were annotated by GO enrichment analysis (http://geneontology.org/)[85–87].

**Gene list identification.** Annotation of TSSs and TESs for 55487 genes are downloaded from Gencode vM22 and active TSSs for mESC identified through START-seq are provided by Dr. Karen Adelman's lab[88]. If active TSS is located within ±1 kb region from the TSS of the 55487 genes, the TSS of this gene is replaced by the position of this active TSS. Genes with length (TSS-TES) < 2 kb and without PRO-seq signal at either promoter-proximal pausing (TSS-100bp + TSS + 300 bp) or early elongating regions (TSS + 300 bp ~ TSS + 2 kbs) were removed to avoid genes with unclear transcripts. Thus, 15265 genes with the length of transcription (TSS-TTS) > 2kb with valid expression are remained for further study.

**ATAC-seq and ChIP-seq.** Data from two biological replicates were first compared ($R^2 > 0.9$), and then merged into a single read file for each time point. ATAC-seq peaks (were then called using MACS2[89] with the following parameters: -q 0.01 –nomodel –shift 75 –extsize 150. To get a union set of peaks from all samples (WT and dAT mutants), MACS2 peaks from each condition were merged using the mergePeaks module from HOMER[90] (default parameters). For identifying differentially accessible regions, the union set of peaks was annotated by Homer and then divided into promoter (−1 kb to +1 kb), and intronic-intergenic regions. Read counts for all peaks in the union set were obtained using the featureCount module of Subread package[91] and differential analysis was done using edgeR[92]. Heatmaps were created using deeptools[83] plotHeatmap function. Data from two biological replicates were first compared ($R^2 > 0.9$), and then merged into a single read file for each time point. ATAC-seq peaks (were then called using MACS2[89] with the following parameters: -q 0.01 –nomodel –shift 75 –extsize 150. To get a union set of peaks from all samples (WT and dAT mutants), MACS2 peaks from each condition were merged using mergePeaks module from HOMER[90] (default parameters). For identifying differentially accessible regions, the union set of peaks was annotated by Homer and then divided into promoter (−1 kb to +1 kb), and intronic-intergenic regions. Read counts for all peaks in the union set were obtained using the featureCount module of Subread package[91] and were differential anlysis was done using edgeR[92]. Heatmaps were created using deeptools[83] plotHeatmap function.

To determine the known motif enrichment findMotifsGenome module of the HOMER package was used[90]. For motif heatmap analysis, differential ATAC-seq peaks between WT and dAT mutants were used to identify the known motifs using 'findMotifsGenome.pl' from Homer. Then, p-value of identified motifs were transformed into Z-score and plotted as a heatmaps using the R ggplot package. For ATAC footprint analysis, normalized ATAC files were corrected using 'ATACorrect' module from TOBIAS[45]. Next, the average ATAC-signals were calculated (around +/− 100 bp of the center of the motif enriched peaks) and plotted using plotProfile from deeptools.

For ChIP-seq analysis, Paired-end 75 bp reads were aligned to mm10 using Bowtie 2[78] alignment tool using –very-sensitive-local preset parameter. Data from two biological replicates were first compared to check for concordance ($R^2 > 0.9$), and then merged into a single read file for each cell type for further downstream analysis. The high confidence peak sets were selected from biological replicates using the intersectBed function from BEDTools[80] with default parameters. For histone and Brg1 ChIP, peaks were called using MACS2[89] with the following parameters: –broad -q 0.05 –nomodel -extsize 500. To compare the signals between IP and input, 'bdgcmp' from MACS2 option was used. Peaks were annotated using the annotatePeaks module of HOMER package[90]. All the heatmaps were drawn using 'plotHeatmap' from deeptools[83]. For ChIP-seq analysis, Paired-end 75 bp reads were aligned to mm10 using Bowtie 2[78] alignment tool using –very-sensitive-local preset parameter. Data from two biological replicates were first compared to check for concordance ($R^2 > 0.9$), and then merged into a single read file for each cell type for further downstream analysis. The high-confidence peak sets were selected from biological replicates using the intersectBed function from BEDTools[80] with default parameters. For histone and Brg1 ChIP, peaks were called using MACS2[89] with the following parameters: –broad -q 0.05 –nomodel -extsize 500. To compare the signals between IP and input, 'bdgcmp' from MACS2 option was used. Peaks were annotated using the annotatePeaks module of HOMER package[90]. All the heatmaps were drawn using 'plotHeatmap' from deeptools[83].

**Mass spectrometry.** Following data acquisition, the data was searched using an empirically corrected library and a quantitative analysis was performed to obtain a comprehensive proteomic profile. Proteins were identified and quantified using EncyclopeDIA and visualized with Scaffold DIA using 1% false discovery thresholds at both the protein and peptide level[93]. The UniProt database for Mus musculus was used for the database search. Protein exclusive intensity values were assessed for quality using ProteiNorm, a user-friendly tool for a systematic evaluation of normalization methods, imputation of missing values and comparisons of different differential abundance methods[94]. Popular normalization methods were evaluated including log2 normalization (Log2), median normalization (Median), mean normalization (Mean), variance stabilizing normalization[95,96], quantile normalization (Quantile), Cyclic loess normalization (Cyclic Loess), global robust linear regression normalization (RLR), and global intensity normalization (Global Intensity). The individual performance of each method was evaluated by comparing of the following metrices: total intensity, Pooled intragroup Coefficient of Variation (PCV), Pooled intragroup Median Absolute Deviation (PMAD), Pooled intragroup estimate of variance (PEV), intragroup correlation, sample correlation heatmap (Pearson), and log2-ratio distributions. The data was normalized using Cyclic Loess and statistical analysis was performed using Linear Models for Microarray Data (limma) with empirical Bayes (eBayes) smoothing to the standard errors[95]. Proteins with an FDR adjusted p-value < 0.05 and a fold change > 2 are considered significant.

**Reporting summary**

Further information on research design is available in the Nature Portfolio Reporting Summary linked to this article.

## Data availability

The CX-MS, ChIP-Seq, PRO-seq, and ATAC-seq data generated in this study have been deposited in the ProteomeXchange and Gene Expression Omnibus with identifier PXD040664 and accession code GSE207793, respectively. These data sets are available and can be obtained from above mentioned public repository using respective accession code. We did not use any custom-made code for the analysis. Source data are provided with this paper.

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

## Acknowledgements

Funding support was provided by the National Institutes of Health (grant R01GM136974 to J. A. R, grant ZIA AG000679 to P.S., and grant R01GM108908 to B.B.). NGS sequencing was performed at the Science Park NGS Core, with support from CPRIT Core Facility Support Grant RP120348. Mass spectrometry was done at the University of Arkansas Medical Center Proteomics Core with support from NIH grant R24GM137786. Laser scanning confocal microscopy and image processing were performed at the MD Anderson Cancer Center Epigenetics & Molecular Carcinogenesis Flow Cytometry and Cell Imaging Core (EMC-FCCIC) with funding support provided by the CPRIT core facility grant RP170628. Collene Jeter provided invaluable support with confocal microscopy. D.S. and S.H. received support from the Center of Cancer Epigenetics at MD Anderson Cancer Center.

## Author contributions

The yeast Snf2 mutant strains were prepared and SWI/SNF complexes purified by P.S. and S.H. and the remodeling and ATPase assays were done by A.H and P.S. The spot assays were done by J. Le. The CX-MS were conducted by S.H., J.P., and J. Lu with assistance from J. A. R. K.F. to prepare the PRO-seq, ATAC-seq, and Brg1 ChIP-seq samples. The H3K4me1, H3K4me3, and H3K27ac samples were prepared by K.F., D.S., and A.J. Cell imaging, qRT-PCR, and Brg1 complex purification experiments were done by D.S. J. Le performed the bioinformatic analysis of PRO-seq and D.S. did the bioinformatic analysis for ChIP-seq, and ATAC-seq. Y.C.L and D.S. made and characterized all the mESC clones with HA tag and knock-in of the AT-hook deletion using CRISPR-Cas9. Y.L. did all the initial processing of NGS data. B.L. assisted and supervised the bioinformatic analysis and B.B. supervised this work. B.B., D.S., J.Le, J.A.R., and P.S. assisted in the manuscript preparation.

## Competing interests

The authors declare no competing interests.
