## [Peer Review File · Nature Communications]

The AT-hook is an evolutionarily conserved auto-regulatory domain of SWI/SNF required for cell lineage primingEditorial Note: Parts of this Peer Review File have been redacted as indicated to maintain the confidentiality of unpublished data.

REVIEWER COMMENTS

Reviewer #1 (Remarks to the Author):

This manuscript describes the *in vitro* role of the AT-hooks in SWI/SNF remodeling in yeast and relates it to *in vivo* function in yeast and to cBAF function in mammalian ES cells. The authors provide nice MS evidence that the AT hook is not required for SWI/SNF or BAF complex formation/stability and instead associates with H3 tails. *In vitro*, the authors find that deletion of the AT hooks significantly affect the kinetics of remodeling but perhaps only slightly affects the affinity of SWI/SNF to DNA (although it does affect the affinity to nucleosomes). In various yeast models of *snf2* dependency, the AT hook is either dispensable or only somewhat affects viability. It has a dramatic phenotype in yeast models of Gcn5 function and nucleosome occupancy. In mouse ESC models, the AT hook deletion results in impaired enhancer activation and state-specific transcription factor binding. The genomics data is clear and indicates that Naïve-specific sites are disrupted upon AT hook deletion while many other BRG1-dependent sites are not. There are a lot of interesting findings about SWI/SNF mechanism and function; however, some of the conclusions would be better supported with additional controls and/or stated with reservations. In addition, some of the writing requires some editing for clarity. I've detailed these concerns below:

Major concerns:

The deletion of the AT hooks doesn't affect complex formation but might affect bromodomain positioning within the complex. Many of the observed effects could be related to impaired bromodomain function in addition to the loss of AT hook function. Especially considering the MS results indicates that AT hooks associate with H3 tails (which could facilitate bromodomain recognition of acetylation) and that the AT hook is particularly important for Gcn5-dependent effects *in vivo*. The authors propose the opposite situation on line 338 in the discussion (bromodomain recruits AT hook to H3 tails) but the converse could also be true. A good control for at least some of the yeast phenotype studies, if not the mammalian studies, would be the bromodomain mutant or deletion. Another would be the deletion of both the AT hooks and bromodomain to see if there is any bromodomain function independent of the AT hooks.

Another thing to consider is whether the sites with decreased TF binding in the AT hook deletion are H3K14Ac marked (if those datasets exist for ESCs)

Many of the interpretations of the comparison between AT hook deletion here and BRG1 deletion in ESCs are based on published work using FBS+LIF and not necessarily the 2i system used here. Many have observed a less stable naïve state using FBS+LIF (described in Nat Cell Biol. 2018 May; 20(5): 565–574. <https://www.ncbi.nlm.nih.gov/pmc/articles/PMC5937285/>) that is more dependent on a variety of chromatin regulators (including BRD4 in this paper). While I am not sure whether it is published, BRG1 is rumored to be more dispensable for pluripotency in 2i than FBS+LIF. The full BRG1 KO or BRG1 CD mutant should be explored in some capacity as a control to support the statement that the AT hook deletion has different roles in enhancer priming than the KO (or ATPase dead). Redoing all the sequencing shouldn't be necessary, but some follow up to define how the AT hook deletion differs from BRG1 deletion/ATPase dead mutant in these conditions would be necessary to support these statements based on the literature. The commercially available ATPase inhibitors and PROTACS might help in this situation, although I'm not sure how effective they are in mouse cells.

Less Major concerns:

In general, the statements connecting *in vitro* activities of the AT hook deletion with yeast directly to all phenotypes observed with the AT hook deletion in ESCs should be tempered slightly to not overstate that all phenotypes are related to a 10x fold decrease in remodeling.

In yeast (Figure 4c) The authors interpret this difference in the MNase-Seq as attenuation from AT-

hook deletion has a bigger effect than SNF2 deletion or ATPase-dead SNF2, but I'm not sure if that is a reasonable interpretation without multiple replicates. Are there controls or follow up experiments that can be performed here to support this conclusion?

Some context and explanation regarding the Gcn5-dependent functions of yeast would be helpful. This seems potentially relevant and perhaps implies a cooperative role with the bromodomain. Does this relate to the finding in 4C that the AT hook is important for chromatin structure? Are those Gcn5 sites? Or does it relate to the roles in ESCs? Has that been defined for Gcn5?

Is H3K4me1 known to require BRG1?

Minor concerns (editing mostly):

First line of abstract: inappropriate semicolon

Line 176: Define whether K798A mutant is the same as snf2-CD used in Figure 4.

Pg. 7 line 192: This statement is confusing: We find the chromatin structure at promoters is altered more when SWI/SNF activity is downregulated ~10-fold than >100-fold or the catalytic subunit is absent. What does this mean? I assume this is comparing the AT hook deletion to SnAC deletion and KO?

S4e: The type of enrichment isn't specified in the legend. Based on the text, it seems to be protein levels from IP-MS? What do the numbers indicate? There is an extra R in Gltsr1. Were peptides from any of the other BAF subunits detected?

Line 217: esBAF is kind of a composite of all three, so if mentioning gBAF it might be worth mentioning PBAF as well. Although I don't know if there is a published ChIP-Seq in ESCs, it's typically at promoters in other cell types.

Line 213: confusing sentence: We observe in the two different states most Brg1 sites are cell state-specific in naïve and primed pluripotent cells with significantly more Brg1 binding detected in the primed state

Line 240: confusing sentence: The stage-specific binding of many TFs depends on Brg1 and its AT-hook with 5,061 and 5,350 accessible sites in respectively the naïve and primed states depending on the AT-hook of Brg1 and more than 7,000 sites that remain unaltered upon deletion of the AT-hook (Figures S5f-g).

Line 243: I think Figure 5e should be SI 5e?

Line 327: "This is a little confusing: we observe when the AT-hook regulates the nucleosome remodeling activity of SWI/SNF it does so outside of promoting binding to DNA or chromatin" but then stating that the DNA-dependent ATPase activity depends on the AT hook, which implies binding to DNA (and chromatin from the nucleosome data). Maybe this needs to be restated to indicate that the AT hooks deletion doesn't affect overall chromatin affinity?

Reviewer #2 (Remarks to the Author):

In this manuscript Saha et al. investigate the functional importance of the BRG1 AT-hook. This is a highly conserved intrinsically disordered region found at the C-terminus of BRG1. Though it has previously been found to associate with DNA, its role in the activity of BRG1 as remained elusive. Here, using a powerful combination of in vitro biochemistry and various cellular assays the authors

convincingly demonstrate that the AT-hook plays an important functional role beyond chromatin association. Rather, it alters the enzymatic activity of SWI/SNF leading to changes in chromatin structure, transcriptional regulation, and cell lineage priming. Overall, these results are exciting and should be of broad interest. I have some concerns regarding the interpretation of data that should be addressed before publication in addition to some more minor concerns throughout.

Major

- 1) The authors note that there is "some interplay between AT-hooks and histones". However, this is entirely unclear and should be expanded. Furthermore, there are many differences between DNA and nucleosomes outside of the histones.
- 2) The authors interpret their CX-MS data to indicate direct contacts. However, cross-links could arise simply from two regions being positioned adjacent to each other without a direct interaction. Thus, the conclusion that the AT-hook and H3 tail directly interacts is not correct and this interpretation must be altered. The authors simply cannot state that these two directly interact from their data. The same holds for other interactions identified from cross-links.

Minor

- 1) The sentence in the abstract "Attenuation of SWI/SNF remodeling activity by the AT-hook..." is misleading. Rather the deletion of the AT-hook attenuates activity not the presence of it. This should be reworded.
- 2) The authors should be consistent in their nomenclature of the deletion mutant as sometimes δ is used other times Δ is used and other times the Greek δ is used.
- 3) In Figure 1a,b – the schematics are a bit confusing. For the reader not as familiar it will not be clear that there is only 1 AT-hook in some species and 2 in the other. This should be made clear. In addition – it is critical to make clear in this figure and in the text which residues were actually deleted.
- 4) For the in vitro assays it would be appreciated if more detail was provided in the text regarding substrates used. The authors talk about DNA and nucleosomes but with no information on what these substrates are. It would be nice not to have to dig through methods/figure legends. In addition, even in the methods it is not clear if the histones are recombinant xenopus or purified from xenopus. The process of producing the histones needs to be described. This is especially important to understand the modification state.
- 5) The authors should be a bit clearer in discussing affinity as they are both measuring K_M and K_D here. Please be sure to distinguish between them better rather than just saying affinity. In addition, it is unclear when they say the K_D is more or less. Rather please indicate if it is increased or decreased or stronger or weaker.
- 6) When discussing something as being significantly different or not please use a statistical test to back this up.

Reviewer #3 (Remarks to the Author):

Snf2 in yeast and its homolog Brg1 in mammalian cells are the catalytic subunit of the SWI/SNF family complexes, which play a major role in regulation of chromatin structure and gene expression. The ATPase domain Snf2/Brg1 is responsible for the remodeling activity. There is increasing interest in which factors and how to regulate the remodeling activity of SWI/SNF. In this manuscript, the authors report that the AT-hook motif regulates the SWI/SNF remodeling activity in vitro, and they validate the importance of the AT-hook motif for the survival of yeast cells, and for lineage priming in mouse ES cells.

An interesting finding of the article is the distinct phenotypes of the AT-hook deletion mutant from those of the complete inactivation of the enzyme. These findings provide more information on the fine tuning of the remodeling activity of SWI/SNF. However, there are several issues that need to be

addressed.

Major points:

1. While this study is rich in phenotypic description of the AT-hook mutant in vitro and in cells, it is short of mechanistic explanation for the importance of the AT-hook motif. The authors approach this question by performing crosslinking-mass spectrometry analysis in Fig. 3b and 3c. However, the interpretation of their findings is less satisfactory. They found that the AT-hook is cross-linked to the H3-tail, and propose that this interaction enhances the nucleosome affinity. Yet, in the discussion section, they seem to disfavor the importance of the H3-AT hook interaction. Validation of the importance of the H3-AT hook interaction is required to provide the missing mechanistic insight.
2. In Fig. 1d, the remodeling activity is not quantified and repeated. Moreover, instead of similar efficiencies as claimed by the authors, the WT complex seems to have a higher activity, as gauged by the band intensities of lanes 3 and 7.
3. In Fig. 1e, the authors claimed that saturating amounts of SWI/SNF is used and the nucleosomes are fully bound. However, in these experiments, the concentration of SWI/SNF used (7.5 nM) is similar to the KD of the WT complex (Fig. 3b, 7.67 nM), and lower than that of the mutant (20.5 nM). Under these conditions, the nucleosomes are not fully bound.
4. In figure 4a, the authors observed that the AT-hook and the SnAc domains are not required for cell viability, and conclude that the physical presence of SWI/SNF is required rather than its chromatin remodeling activity. However, how can the authors exclude the possibility that the residual activity of the mutant complex is enough for the viability?

Minor points :

1. From the main text, "...crosslinking lysine 1441 in the first AT-hook and additional crosslinks between the regions flanking the AT-hooks and SnAC domain (Figure 3e)." (page 6, line 165), however, lysine 1314 in the SnAC domain links lysine 1444 in the AT-hook domain in Figure 3e. So, which residue in the AT-hook domain crosslinks K1314 in the SnAC domain? Besides, the raw data of CX-MS should be provided as supplementary information.
2. In the main text, "These accessible sites reflect binding of stage-specific TFs and motif analysis reveals the pluripotency TFs Oct4, Nanog, Klf family and Sox2 are most enriched at these sites that are naïve-specific and the epiblast-specific TFs Zic2, Zic3, Otx2 and Glis3 are most enriched in the primed state (Figure S5c-d)." (page 8, line 234-237), however, Nanog and Sox2 are not shown in Figure S5c, Otx2 and Glis3 are not shown in Figure S5d. Did I miss some information?
3. In the Materials and methods part, it is mentioned that mono/di-nucleosomes were used in ATP assays (page 15, line 425-427). It would be helpful to point out the concentrations of the nucleosomes used, and why di-nucleosomes are used.
4. The x-scale of Figure 2d is inconsistent with Figure 2b.
5. The unit of the ATP concentration is quite confusing. "... ATP that varied from 0.2 to 800 mM ..." (page 29, line 817), however, "µM ATP" shown in the figure. Likewise, "limiting ATP (4.4 mM) (page 28, line 807)", and the unit of "Km" in the Table 1.
6. In Figures 8b-c, the number of independent experiments should be mentioned, error bars significance and p values need to be provided. In Figure 8d, for the data set of ΔAT2, the error bars don't present at the center of columns.
7. In Figure S8f, the ordinate of box plots doesn't show, "ATAC signals"?
8. In Figures 4a and 4b, does "snf2-CD" mean "snf2-catalytic dead"? It should be indicated when shown the first time.
9. There are some mismatches in the main text and legend:
 - λ "ethanol as an energy source and is not required for using raffinose (Figure 4a) ..." (page 6, line 188), this should reference Figure 4b.
 - λ "shows there are two classes of pluripotency TF binding sites (Figure 5e)." (page 8, line 239), this should reference Figure S5e.

λ "... (Figures 7e and S7d). Some of these genes ..." (page 10, line 291), this should reference Figures 7e and S7c.

λ "(e) The Lys-Lys crosslinking pattern between Snf2 and histones is shown for nucleosome" (page 29, line 830), this should be Figure 3c's legend. "(c-d) The Lys- Lys ... crosslinked pairs." (page 29, line 826-829) seems to be the legend of Figures 3d and 3e, it should be re-written clearly.

λ "... H3K27ac at naïve (f) and primed (g) specific enhancers ..." (page 30, line 865), this should reference "naïve (c) and primed (d)"

λ "(a-d) Meta-analysis of PRO-seq signals..." (page 30, line 873), this should be the legend of Figures 7a, 7c, 7e, 7g; "... (e-g) Bar graphs show gene ontology (GO) ..." (page 31, line 877), this should be the legend of Figures 7b, 7d, 7f.

λ "(a-b) Representative ... in Figures 3a and S3b to measure ..." (Figure S1. Legend) this should reference Figures 3a and 3b.

λ "(a-b) Representative ... (a) the relative affinity of WT and Δ AT SWI/SNF for nucleosomes ..." (Figure S1. Legend), the Figure S1a shows the binding of WT and Δ AT SWI/SNF for DNA, not for nucleosomes.

10. There are a few spelling mistakes:

λ "Brg1 and its AT-hook are required to activate transcription in the naïve and primed" (page 9, line 246) is an incomplete sentence, it should be "Brg1 and its AT-hook are required to activate transcription in the naïve and primed state."

λ "Brg1 and it AT-hook..." (page 11, line 312), "it" should be replaced by "its".

λ "37.35:1" (page 14, line 417), is it "37.5:1"?

λ Us "μl" instead of "μls" (page 15, line 444).

λ "...Candida albicans (C.s.) ..." (page 29, line 799), "(C.s.)" should be corrected as "(C.a.)".

λ "Lys-ys" (page 29, line 829) should be corrected as "Lys-Lys".

λ For the description of the temperature, use "°C" instead of "C". (line 403, line 414, line 419, line 437, and line 475)

λ "BS3" as the acronym of "amine-reactive crosslinker bis[sulfosuccinimidyl] suberate" used in the whole text, "BS3" is used in line 444 and line 450 on page 15, they are better replaced by "BS3".

11. There are some repeat editing needed to delete:

λ Line 465-471 is a copy of line 432-438.

λ Line 669-682 is a copy of line 655-668.

λ Line 705-714 is a copy of line 696-705.

λ Line 732-742 is a copy of line 723-732.

λ Line 1125-1132 is a copy of line 784-792.

12. In the References part, there are some references not present correctly, line 960, line 1053, line 1061, and line 1075.

13. Please show clearly which residues are deleted in Δ AT mutation of yeast snf2 and Δ AT mutation of mouse Brg1.

14. Throughout the manuscript, there are many awkward sentences that do not express the meaning clearly or correctly. For instance,

λ We observe the AT-hook however regulates the intrinsic DNA-stimulated ATPase activity without promoting SWI/SNF recruitment to DNA or nucleosomes by increasing the reaction velocity a factor of 13 with no accompanying change in substrate affinity (KM). --This statement in the abstract contradicts the observed difference in the nucleosome binding affinity.

λ SWI/SNF binding to nucleosomes increases the binding affinity of WT SWI/SNF for substrate over 3-times that with DNA (311 versus 98 nM)

λ The stage-specific binding of many TFs depends on Brg1 and its AT-hook with 5,061 and 5,350 accessible sites in respectively the naïve and primed states depending on the AT-hook of Brg1 and more than 7,000 sites that remain unaltered upon deletion of the AT-hook.

λ Similarly, there is a group of 1,003 genes in Δ AT1 (720 genes in Δ AT2) that are up regulated in the primed versus the naïve state that fails to be activated in the AT mutant (Figure 7c and S7b).

Reviewer #4 (Remarks to the Author):

The publication by Saha et al. aims at understanding the molecular function of the AT-hook domain of the SWI/SNF chromatin remodeler. Despite its conservation across eukaryotes, the function of the AT-hook is poorly characterized. The publication starts with what seems to me a nice in vitro description of the AT-hook function. They find that the AT-hook enhances both ATPase and remodeling/sliding activities without affecting the DNA affinity of SWI/SNF. They then map by CX-MS the interaction of yeast Snf2 with nucleosomes showing an interaction with the N-terminal tail of H3. They move then to the in vivo part. First in *S. cerevisiae*, by testing the importance of the AT-hook under different growth conditions but also by assessing nucleosome organization at promoters. Afterwards, the authors analyze the importance of the AT-hook in cell lineage priming in mESCs concluding that this domain is required for transcription activation in both naïve and primed states. In conclusion, the authors stress the point that studying SWI/SNF through deletion of the AT-hook gives a more balanced pattern than deletion or inhibition of the catalytic subunit.

This publication represents a massive amount of work. However, if I feel completely convinced by the in vitro part, I feel that the in vivo section is more confused. Nevertheless, considering the novelty in studying the AT-hook domain and the growing interest for CRs during differentiation, I think that this study could be of broad interest after major revisions.

Major comments:

- 1) For the yeast section, it seems surprising that *snf2* delta does not affect -1/+1 nucleosomes positioning. Indeed, it has been shown that *snf2* delta (or depletion) leads to a shrinkage of the Nucleosome-Depleted Region (Rawal et al., 2018, Genes&Dev; Kubik et al., 2019, NSMB) mainly for highly expressed genes, the ones with wider NDRs. However, this is something that is not detected in this experiment. Moreover, the deletion of SNF2 has a weaker phenotype than the AT-hook deletion. This does not correlate with the growth. Either there is an issue with the MNase-seq experiment or the nucleosome organization at promoters does not induce the growth phenotype. I am more in favor of problem with the MNase treatment. Did you test different concentrations of MNase? Could you provide the metagene plots not directly as a ratio over the wt condition? Instead of performing the experiment in YPD, phenotypes might be more convincing in SM or Raffinose conditions?
- 2) For the mESCs section, Figures 5C and E are very striking. However, I think it could be a control to perform the ChIP-seq of at least one naïve TF (Oct4 for example) and one primed factor (Zic3). I agree that the TOBIAS framework is validated but it is still an indirect approach. Is it possible to provide some statistics on the DNA footprint analysis?
- 3) Statistics have to be provided for Figure 8B and C.
- 4) In general, it would be nice to have the metagene plots at the top of heatmaps. Sometimes, differences in colors are difficult to catch at first (at least, for me).
- 5) A more detailed introduction would be appreciated to give more background about CRs and the results already obtained with mESCs.
- 6) Figures 7A, C, E and G have to be better labeled with x and y axes.
- 7) Line 28: "Similarly, the AT-hook is required in yeast SWI/SNF for activation of genes". The growth is tested but not the gene expression. It affects growth suggesting that...
Line 260: "these data demonstrate that the catalytic activity of Brg1" is again a shortcut. Since the AT-hook regulates catalytic activity, it suggests that...
- 8) Line 24: "Attenuation of SWI/SNF remodeling activity by the AT-hook". If I understood well, the AT-hook enhances (not attenuates) SWI/SNF activity. If I am not wrong, I am not sure that the word "attenuation" along the publication is the good word.
- 9) Title: can we really say "auto-regulatory"? The AT-hook is a domain that enhances remodeling, but does it regulate itself?

I am sure that the mESCs part can be lighten and making it easier to read. I do not criticize the quality of the data which seems excellent. To give an example, Figure 5A can be switched to the Supp since it is quite close to the Figure 5B (but only for intronic and intragenic peaks). On the contrary, Figure S5F is very striking and might be moved to the main figures. In my opinion, the authors can rearrange a bit this part to make it more user-friendly.

Reviewer #1

(1) This manuscript describes the *in vitro* role of the AT-hooks in SWI/SNF remodeling in yeast and relates it to *in vivo* function in yeast and to cBAF function in mammalian ES cells. The authors provide nice MS evidence that the AT hook is not required for SWI/SNF or BAF complex formation/stability and instead associates with H3 tails. *In vitro*, the authors find that deletion of the AT hooks significantly affect the kinetics of remodeling but perhaps only slightly affects the affinity of SWI/SNF to DNA (although it does affect the affinity to nucleosomes). In various yeast models of *snf2* dependency, the AT hook is either dispensable or only somewhat affects viability. It has a dramatic phenotype in yeast models of *Gcn5* function and nucleosome occupancy. In mouse ESC models, the AT hook deletion results in impaired enhancer activation and state-specific transcription factor binding. The genomics data is clear and indicates that Naïve-specific sites are disrupted upon AT hook deletion while many other BRG1-dependent sites are not. There are a lot of interesting findings about SWI/SNF mechanism and function;
- We appreciate the positive comments of the reviewer.

(2). The deletion of the AT hooks doesn't affect complex formation but might affect bromodomain positioning within the complex. Many of the observed effects could be related to impaired bromodomain function in addition to the loss of AT hook function. Especially considering the MS results indicates that AT hooks associate with H3 tails (which could facilitate bromodomain recognition of acetylation) and that the AT hook is particularly important for *Gcn5*-dependent effects *in vivo*. The authors propose the opposite situation on line 338 in the discussion (bromodomain recruits AT hook to H3 tails) but the converse could also be true. A good control for at least some of the yeast phenotype studies, if not the mammalian studies, would be the bromodomain mutant or deletion. Another would be the deletion of both the AT hooks and bromodomain to see if there is any bromodomain function independent of the AT hooks.

- Based on a previous study from Jerry Workman's lab (Hassan AH, et al. Cell. 2002;111(3):369-79. PubMed PMID: 12419247), the AT-hook deletion strain does not behave the same as deletion of the bromodomain. While the AT-hook deletion mutant has sulfometuron methyl (SM) phenotype, deletion of the bromodomain of *Snf2* does not (see figure below from this paper) which is discussed in the main text. The discussion part now reads as follows: "The AT-hook has a modest secondary role in enhancing SWI/SNF affinity for nucleosomes through its interactions with the N-terminal tail of histone H3 and raises the question as to whether the AT-hook dependent effects could be due in part to perturbation of the bromodomain⁶². We think this possibility is unlikely given that when the bromodomain is deleted there is no change in the *in vitro* remodeling activity of SWI/SNF with recombinant nucleosomes lacking modifications and is quite different than when the AT-hook is deleted³². The same is also true for SWI/SNF activity *in vivo* as shown by deletion of the bromodomain of *SNF2* having no adverse phenotype when amino acid biosynthesis is inhibited by sulfometuron methyl in contrast to that observed when the AT-hook is deleted³⁵. These last data suggest that if the AT-hook interactions with the H3 tail have a role *in vivo* then it is likely independent of the bromodomain, consistent with the interactions of the AT-hook and H3 tail being direct as suggested by the protein crosslinking data."

(3). Another thing to consider is whether the sites with decreased TF binding in the AT hook deletion are H3K14Ac marked (if those datasets exist for ESCs).

- The H3K14ac mark in mouse embryonic stem cells is localized to promoters, including bivalent and inactive promoters, and enhancers and correlates with CpG content (Karmodiya, K et al., BMC Genomics 2012; 13:424 PMID: 22920947). Given the effects we observe are enhancer specific, there is not a likely connection between where H3K14ac occurs and the effects we observed when the AT-hook is deleted in mESCs.

(4). Many of the interpretations of the comparison between AT hook deletion here and BRG1 deletion in ESCs are based on published work using FBS+LIF and not necessarily the 2i system used here. Many have observed a less stable naïve state using FBS+LIF (described in Nat Cell Biol. 2018 May; 20(5): 565–574. <https://www.ncbi.nlm.nih.gov/pmc/articles/PMC5937285/>) that is more dependent on a variety of chromatin regulators (including BRD4 in this paper). While I am not sure whether it is published, BRG1 is rumored to be more dispensable for pluripotency in 2i than FBS+LIF. The full BRG1 KO or BRG1 CD mutant should be explored in some capacity as a control to support the statement that the AT hook deletion has different roles in enhancer priming than the KO (or ATPase dead). Redoing all the sequencing shouldn't be necessary, but some follow up to define how the AT hook deletion differs from BRG1 deletion/ATPase dead mutant in these conditions would be necessary to support these statements based on the literature. The commercially available ATPase inhibitors and PROTACS might help in this situation, although I'm not sure how effective they are in mouse cells.

- There is a recent report in BioRxiv (<https://doi.org/10.1101/2023.03.07.531379>) from the lab of Karen Adelman that have used BRM014, a chemical inhibitor that blocks the catalytic activity of Brg1, in naïve mESCs. BRM014 interferes with stem cell proliferation and cell morphology that show Brg1 catalytic activity is required for stem cell maintenance. We show below how the transcriptional profile with BRM014 compare to ours with the ΔAT mutant in the naïve stage and demonstrate how extensively BRM104 effects transcription. The differences between the two are quite extensive as can be seen and illustrates the points we made both in yeast and mESCs. The data here presented is not include in our manuscript because the BRM014 data is in a paper that is currently under review at another journal and is not publically available.

[REDACTED]

(5) In general, the statements connecting in vitro activities of the AT hook deletion with yeast directly to all phenotypes observed with the AT hook deletion in ESCs should be tempered slightly to not overstate that all phenotypes are related to a 10x fold decrease in remodeling.

- We completely agree that the phenotype could be influenced by other factors and did not intend to apply that it is the only potential reason for the observed phenotypes. We have revised our text accordingly.

(6) In yeast (Figure 4c) The authors interpret this difference in the MNase-Seq as attenuation from AT-hook deletion has a bigger effect than SNF2 deletion or ATPase-dead SNF2, but I'm not sure if that is a reasonable interpretation without multiple replicates. Are there controls or follow up experiments that can be performed here to support this conclusion?

- We understand this question as to whether we have done sufficient biological replicates to ensure the reproducibility and statistical significance. There are two independent biological replicates and as stated in the methods section we have done the necessary statistical analysis to ensure the validity of our results. There are several controls as we not only compare Δ AT to WT, but also to the catalytical dead version of Snf2 and the SnAC domain deletion.

(7) Some context and explanation regarding the Gcn5-dependent functions of yeast would be helpful. This seems potentially relevant and perhaps implies a cooperative role with the bromodomain. Does this relate to the finding in 4C that the AT hook is important for chromatin structure? Are those Gcn5 sites? Or does it relate to the roles in ESCs? Has that been defined for Gcn5?

- In yeast, subunits of the SAGA complex have been found to promote SWI/SNF binding to Gcn4 bound sites, independent of the histone acetyltransferase activity of Gcn5 (Yoon S. et al., Mol Cell Biol 2003 23(23):8829-9945). There are multiple lines of evidence that Gcn4 recruits SWI/SNF to DNA both in vivo and in vitro and Gcn4 can bind independently of SWI/SNF. Based on these data there is no reason to suggest that the bromodomain of Snf2 is involved in the effect of SWI/SNF on Gcn4 activation, consistent with lack of a phenotype observed when the bromodomain of Snf2 is deleted and sulfometuron methyl is added (Hassan AH, et al. Cell. 2002;111(3):369-79. PubMed PMID: 12419247). These references are discussed in the main text. It is also important to remember the MNase-seq experiments are done with yeast grown

in YPD and there is no selection for SWI/SNF in terms of Gcn5 and these data do not reflect any connection to Gcn5.

(8) Is H3K4me1 known to require BRG1?

- The MLL3/4 complex is the enzyme primarily responsible for monomethylating lysine 4 of H3 (H3K4me1) and MLL3/4 and Brg1 have been shown to reciprocally enhance their binding in vivo (Park Y.K. et al., Nat Commun 2021; 12:1630. PMID PMC7955098) and the BAF or SWI/SNF complex binds to H3K4me1 and H3K4me1 promotes more efficient remodeling by BAF (Local, A. et al. Nat.Genet 2018; 50(1):73-82. PMID: **29255264**). Both of the references are cited in the paper.

(9) First line of abstract: inappropriate semicolon

- We have corrected the error in the main text.

(10) Line 176: Define whether K798A mutant is the same as snf2-CD used in Figure 4.

- Snf2 catalytic dead mutant is the same as the K798A mutant identified and characterized by Peterson C L, 1996, NAR. We have changed our nomenclature to avoid any confusion.

(11) Pg. 7 line 192: This statement is confusing: We find the chromatin structure at promoters is altered more when SWI/SNF activity is downregulated ~10-fold than >100-fold or the catalytic subunit is absent. What does this mean? I assume this is comparing the AT hook deletion to SnAC deletion and KO?

- Here we are comparing the AT-hook deletion to the catalytically dead Snf2 mutant and have removed this statement to make it less confusing.

(12) S4e: The type of enrichment isn't specified in the legend. Based on the text, it seems to be protein levels from IP-MS? What do the numbers indicate? There is an extra R in Gltscr1. Were peptides from any of the other BAF subunits detected?

- We have added all the information in the revised figure legends. The revised figure legend is as follows: "The table shows the average enrichment of es-BAF and gBAF components in WT and AT-hook deleted mutant cells detected by mass spectrometry in both naïve and primed states. The numbers indicate the normalized average enrichment of peptides of respective proteins and ± represents the standard error of two replicates. Enrichment factor normalization was done using Brg1."

(13) Line 217: esBAF is kind of a composite of all three, so if mentioning gGBAF it might be worth mentioning PBAF as well. Although I don't know if there is a published ChIP-Seq in ESCs, it's typically at promoters in other cell types.

- In mouse ES cells, the expression of PBRM1 (subunit unique to the PBAF complex) is very low at early stages (Ho et al., 2009) and hence, there is no published data available for PBAF at naïve and primed stages of mESCs. Also, esBAF is not a composite of all three but is distinct from GBAF or ncBAF and PBAF.

(14) Line 213: confusing sentence: We observe in the two different states most Brg1 sites are cell state-specific in naïve and primed pluripotent cells with significantly more Brg1 binding detected in the primed state.

- We now modified the sentence as suggested. The revised sentence is as follow- “We observe Brg1 binds sites unique to either the naïve or primed stages (Figures 5a and S4a). At those sites where Brg1 is bound in both the naïve and primed stages, we observe a higher enrichment of Brg1 in the primed state.”

(15) Line 240: confusing sentence: The stage-specific binding of many TFs depends on Brg1 and its AT-hook with 5,061 and 5,350 accessible sites in respectively the naïve and primed states depending on the AT-hook of Brg1 and more than 7,000 sites that remain unaltered upon deletion of the AT-hook (Figures S5f-g).

- We have modified this sentence as follows: “As detected by ATAC-seq, binding of transcription factors at more than 5,000 stage-specific sites depend on the AT-hook of Brg1 either in naïve or primed stages (Figures S5f). At about 7,000 other sites in either naïve or primed stages transcription factor binding is retained when the AT-hook is deleted (Figure S5g).”

(16) Line 243: I think Figure 5e should be SI 5e?

- We have corrected the figure number.

(17) Line 327: “This is a little confusing: we observe when the AT-hook regulates the nucleosome remodeling activity of SWI/SNF it does so outside of promoting binding to DNA or chromatin” but then stating that the DNA-dependent ATPase activity depends on the AT hook, which implies binding to DNA (and chromatin from the nucleosome data). Maybe this needs to be restated to indicate that the AT hooks deletion doesn’t affect overall chromatin affinity? We have changed this to the following: “We observe the AT-hook regulates the nucleosome remodeling activity independent of facilitating SWI/SNF recruitment to nucleosomes, which is different than originally proposed for chromatin modifying proteins containing an AT-hook.”

Reviewer #2

(1). In this manuscript Saha et al. investigate the functional importance of the BRG1 AT-hook. This is a highly conserved intrinsically disordered region found at the C-terminus of BRG1. Though it has previously been found to associate with DNA, its role in the activity of BRG1 as remained elusive. Here, using a powerful combination of in vitro biochemistry and various cellular assays the authors convincingly demonstrate that the AT-hook plays an important functional role beyond chromatin association. Rather, it alters the enzymatic activity of SWI/SNF leading to changes in chromatin structure, transcriptional regulation, and cell lineage priming. Overall, these results are exciting and should be broad interest.

- We appreciate the very positive comments of the reviewer.

(2) The authors note that there is “some interplay between AT-hooks and histones”. However, this is entirely unclear and should be expanded. Furthermore, there are many differences between DNA and nucleosomes outside of the histones. The authors interpret their CX-MS data to indicate direct contacts. However, cross-links could arise simply from two regions being positioned adjacent to each other without a direct interaction. Thus, the conclusion that the AT-hook and H3 tail directly interacts is not correct and this interpretation must be altered. The authors simply cannot state that these two directly interact from their data. The same holds for other interactions identified from cross-links.

- We have revised the text such that instead of stating they interact we say “associate with” based on their proximity. The crosslinking data does however generally correlate well with protein interactions observed in the cryoEM structure. We have changed the statement including “some interplay between AT-hooks and histones” to “In summary, the reduction in ATP hydrolysis that occurs when the AT-hook is deleted is not histone dependent, but the AT-hook likely interacts with histones due to it increasing SWI/SNF affinity for nucleosomes and not for DNA”. The reviewer is correct that there are other difference in inter- and intra-crosslinks of SWI/SNF between free and nucleosome bound that we don’t discuss because we focus on the AT-hook and the parts of SWI/SNF that contact the nucleosome for the purpose of this paper.

(3) The sentence in the abstract “Attenuation of SWI/SNF remodeling activity by the AT-hook...” is misleading. Rather the deletion of the AT-hook attenuates activity not the presence of it. This should be reworded.

- We have revised the sentence as follows: “The catalytic subunit’s AT-hook is required *vivo* for SWI/SNF remodeling activity in yeast and mouse embryonic stem cells.”

(4) The authors should be consistent in their nomenclature of the deletion mutant as sometimes δ is used other times Δ is used and other times the Greek δ is used.

- We have made the requested changes

(5) In Figure 1a,b – the schematics are a bit confusing. For the reader not as familiar it will not be clear that there is only 1 AT-hook in some species and 2 in the other. This should be made clear. In addition – it is critical to make clear in this figure and in the text which residues were actually deleted.

- We have added the following sentence to the figure legend for Figure 1 : “The two AT-hooks of yeast Snf2 was removed by deleting residues 1443-1539 in yeast Snf2” and the main text section including that residues 1401-1423 of Brg1 were removed.

(6) For the *in vitro* assays it would be appreciated if more detail was provided in the text regarding substrates used. The authors talk about DNA and nucleosomes but with no information on what these substrates are. It would be nice not to have to dig through methods/figure legends. In addition, even in the methods it is not clear if the histones are recombinant xenopus or purified from xenopus. The process of producing the histones needs to be described. This is especially important to understand the modification state.

- We clarify the histones were recombinant and provide the appropriate reference. We have also now added the information into the main text as requested.

(7) The authors should be a bit clearer in discussing affinity as they are both measuring K_M and K_D here. Please be sure to distinguish between them better rather than just saying affinity. In addition, it is unclear when they say the K_D is more or less. Rather please indicate if it is increased or decreased or stronger or weaker.

- When we introduce K_M it is indicated to come from the Michaelis-Menten analysis and is defined as substrate affinity – not specifically affinity for nucleosomes or DNA as in K_D . Also, when we first mention K_D , we also refer to this as the Dissociation Constant and that it is obtained from EMSAs. We have used the terms increased and decreased as requested.

(8) When discussing something as being significantly different or not please use a statistical test to back this up.

- We have removed the word significantly, except where there is an accompanying statistical analysis.

Reviewer #3

(1) Snf2 in yeast and its homolog Brg1 in mammalian cells are the catalytic subunit of the SWI/SNF family complexes, which play a major role in regulation of chromatin structure and gene expression. The ATPase domain Snf2/Brg1 is responsible for the remodeling activity. There is increasing interest in which factors and how to regulate the remodeling activity of SWI/SNF. In this manuscript, the authors report that the AT-hook motif regulates the SWI/SNF remodeling activity in vitro, and they validate the importance of the AT-hook motif for the survival of yeast cells, and for lineage priming in mouse ES cells.

An interesting finding of the article is the distinct phenotypes of the AT-hook deletion mutant from those of the complete inactivation of the enzyme. These findings provide more information on the fine tuning of the remodeling activity of SWI/SNF.

- We appreciate the very positive comments of the reviewer.

(2) While this study is rich in phenotypic description of the AT-hook mutant in vitro and in cells, it is short of mechanistic explanation for the importance of the AT-hook motif. The authors approach this question by performing crosslinking-mass spectrometry analysis in Fig. 3b and 3c. However, the interpretation of their findings is less satisfactory. They found that the AT-hook is cross-linked to the H3-tail, and propose that this interaction enhances the nucleosome affinity. Yet, in the discussion section, they seem to disfavor the importance of the H3-AT hook interaction. Validation of the importance of the H3-AT hook interaction is required to provide the missing mechanistic insight.

- While we suspect the AT-hook association with the H3 N-terminal tail may have a function in vivo that involves histone modifications, the modulation of SWI/SNF activity we observe in vitro does not involve posttranslational modified nucleosomes. We find the ATPase activity of

SWI/SNF stimulated by free DNA is reduced 13-fold when the AT-hook is deleted, and thus is a histone independent effect. The same downregulation is observed with nucleosomes, indicating there is no further reduction in catalytic activity when histones are involved. When we compensate for the reduced affinity of SWI/SNF for nucleosomes caused by deletion of the AT-hook, we still observe a reduction in ATPase activity and nucleosome remodeling that are equivalent. These data indicate the interactions of the AT-hook with histones is not involved in the catalytic and remodeling defects found in this study when the AT-hook is removed. There are no further mechanistic details that we can reveal at this time.

(3) In Fig. 1d, the remodeling activity is not quantified and repeated. Moreover, instead of similar efficiencies as claimed by the authors, the WT complex seems to have a higher activity, as gauged by the band intensities of lands 3 and 7.

-We have removed this figure and added another figure in its place – see (4).

(4) In Fig. 1e, the authors claimed that saturating amounts of SWI/SNF is used and the nucleosomes are fully bound. However, in these experiments, the concentration of SWI/SNF used (7.5 nM) is similar to the KD of the WT complex (Fig. 3b, 7.67 nM), and lower than that of the mutant (20.5 nM). Under these conditions, the nucleosomes are not fully bound.

- Figure 1d has been replaced with one that shows nucleosomes are fully bound by Δ AT SWI/SNF (see lane 17).

(5) In figure 4a, the authors observed that the AT-hook and the SnAc domains are not required for cell viability, and conclude that the physical presence of SWI/SNF is required rather than its chromatin remodeling activity. However, how can the authors exclude the possibility that the residual activity of the mutant complex is enough for the viability?

- To avoid confusion we have removed this statement and part of Figure 4a as it detracted from the important message of the phenotypic growth assays.

(6) From the main text, “...crosslinking lysine 1441 in the first AT-hook and additional crosslinks between the regions flanking the AT-hooks and SnAC domain (Figure 3e).” (page 6, line165), however, lysine 1314 in the SnAC domain link lysine 1444 in the AT-hook domain in Figure 3e. So, which residue in the AT-hook domain crosslinks K1314 in the SnAC domain? Besides, the raw data of CX-MS should be provided as supplementary information.

- We have now included a link to the raw CX-MS data. We have removed the last part of the sentence –“and additional crosslinks between the regions flanking the AT-hooks and SnAC domain” – as it was confusing.

(7) In the main text, “These accessible sites reflect binding of stage-specific TFs and motif analysis reveals the pluripotency TFs Oct4, Nanog, Klf family and Sox2 are most enriched at these sites that are naïve-specific and the epiblast-specific TFs Zic2, Zic3, Otx2 and Glis3 are most enriched in the primed state (Figure S5c-d).” (page 8, line 234-237), however, Nanog and Sox2 are not shown in Figure S5c, Otx2 and Glis3 are not shown in Figure S5d. Did I miss some information?

- In the Figure S5c, the motif OTSN is a composite motif that consists of Oct4, Tcf, Nanog and Sox2. We have observed enrichment of Sox2 and Nanog alone with the p-value 1e-239 and 1e-49 respectively which is lower than OSTN (p-value 1e-794). In the Figure S5d, we are showing the motifs having the highest p-value only. However, the p-value of primed motifs Glis3 and Otx2 are 1e-62 and 1e-24 respectively. We have now added these two motifs in the primed group.

(8) In the Materials and methods part, it is mentioned that mono/di-nucleosomes were used in ATP assays (page 15, line 425-427). It would be helpful to point out the concentrations of the nucleosomes used, and why di-nucleosome are used.

- We have corrected the mistakes in this section which involved deleting any mention of dinucleosomes. Concentrations of nucleosomes used are given in the figure legends as they vary depending on the exact assay being used.

(9) The x-scale of Figure 2d is inconsistent with Figure 2b.

- The units used in both figures are the same. The reason for the range however being different is because Figure 2d is a blown-up version of the region shown in Figure 2b by a box with a dotted line.

(10). The unit of the ATP concentration is quite confusing. "... ATP that varied from 0.2 to 800 mM ..." (page 29, line 817), however, "μM ATP" shown in the figure. Likewise, "limiting ATP (4.4 mM) (page 28, line 807)", and the unit of "Km" in the Table1.

- The final concentration of ATP in the reaction was 0.2 to 800 nM and in the figure it is the rate of ATP hydrolyzed that is on the y-axis and is μM ATP hydrolyzed per minute, which is correct. We have corrected limiting ATP to be 4.4 μM instead of mM and K_M to be nM instead of nm.

(11). In Figures 8b-c, the number of independent experiments should be mentioned, error bars significance and p values need to provide. In Figure 8d, for the data set of Δ AT2, the error bars don't present at the center of columns.

- The experimental details are now added to the revised figure legend. In Figure 8d, the inadvertent error occurred during final preparation of the figure. We have corrected the error bars on the revised figure.

(12) In Figure S8f, the ordinate of box plots doesn't show, "ATAC signals"?

- The figure has been corrected

.

(13) In Figures 4a and 4b, does "snf2-CD" mean "snf2-catalytic dead"? It should be indicated when shown the first time.

- Yes, Snf2-CD refers to snf2-catalytic dead. This is now explained in the main text and the label changed to K798A.

(14) There are some mismatches in the main text and legend:

"ethanol as an energy source and is not required for using raffinose (Figure 4a) ..." (page 6, line 188), this should reference Figure 4b.

- Correction made

“shows there are two classes of pluripotency TF binding sites (Figure 5e).” (page 8, line 239), this should reference Figure S5e.

- We now corrected this error in the revised version of the manuscript.

“... (Figures 7e and S7d). Some of these genes ...” (page 10, line 291), this should reference Figures 7e and S7c.

- The figure number has been corrected.

“(e) The Lys-Lys crosslinking pattern between Snf2 and histones is shown for nucleosome” (page 29, line 830), this should be Figure 3c’s legend. “(c-d) The Lys- Lys ... crosslinked pairs.” (page 29, line 826-829) seems to be the legend of Figures 3d and 3e, it should be re-written clearly.

-The text has been revised to “(c-e) The Lys-Lys crosslinking patterns between (c) Snf2 and histones, (d) the C-lobe of the ATPase and SnAC domain and (e) between the SnAC domain and the AT-hooks are shown for nucleosome-bound SWI/SNF. The open circles indicate the positions of the crosslinked lysines and the colored lines show the Lys-Lys crosslinked pairs.”

“... H3K27ac at naïve (f) and primed (g) specific enhancers ...” (page 30, line 865), this should reference “naïve (c) and primed (d)”

- These figures number is now corrected in the revised version.

“(a-d) Meta-analysis of PRO-seq signals...” (page 30, line 873), this should be the legend of Figures 7a, 7c, 7e, 7g; “... (e-g) Bar graphs show gene ontology (GO) ...” (page 31, line 877), this should be the legend of Figures 7b, 7d, 7f.

- We now revised the figures and numbers

“(a-b) Representative ... in Figures 3a and S3b to measure ...” (Figure S1. Legend) this should reference Figures 3a and 3b.

- This error is now corrected in the figure legend.

“(a-b) Representative ... (a) the relative affinity of WT and Δ AT SWI/SNF for nucleosomes ...” (Figure S1. Legend), the Figure S1a shows the binding of WT and Δ AT SWI/SNF for DNA, not for nucleosomes.

- The correction has been made

(15) There are a few spelling mistakes:

“Brg1 and its AT-hook are required to activate transcription in the naïve and primed” (page 9, line 246) is an incomplete sentence, it should be “Brg1 and its AT-hook are required to activate transcription in the naïve and primed state.”

- The correction has been made.

“Brg1 and its AT-hook...” (page 11, line 312), “it” should be replaced by “its”.
“37.35:1” (page 14, line 417), is it “37.5:1”?

- We have made the correction in the revised manuscript.

Use “ μ l” instead of “ μ ls” (page 15, line 444).

“...Candida albicans (C.s.) ...” (page 29, line 799), “(C.s.)” should be corrected as “(C.a.)”.

“Lys-ys” (page 29, line 829) should be corrected as “Lys-Lys”.

For the description of the temperature, use “°C” instead of “C”. (line 403, line 414, line 419, line 437, and line 475)

“BS3” as the acronym of “amine-reactive crosslinker bis[sulfosuccinimidyl] suberate” used in the whole text, “BS3” is used in line 444 and line 450 on page 15, they are better replaced by “BS3”.

- We have made all the changes suggested by reviewer in the revised version of the manuscript.

(16) There are some repeat editing needed to delete:

Line 465-471 is a copy of line 432-438.

Line 669-682 is a copy of line 655-668.

Line 705-714 is a copy of line 696-705.

Line 732-742 is a copy of line 723-732.

Line 1125-1132 is a copy of line 784-792.

- We have deleted all the duplicated lines in the revised manuscript.

(17) In the References part, there are some references not present correctly, line 960, line 1053, line 1061, and line 1075.

- Corrections have been made

(18) Please show clearly which residues are deleted in Δ AT mutation of yeast snf2 and Δ AT mutation of mouse Brg1.

- These details have been added to the main text and for yeast Snf2 is also shown in Figure 1.

(19) Throughout the manuscript, there are many awkward sentences that do not express the meaning clearly or correctly. For instance,

We observe the AT-hook however regulates the intrinsic DNA-stimulated ATPase activity without promoting SWI/SNF recruitment to DNA or nucleosomes by increasing the reaction velocity a factor of 13 with no accompanying change in substrate affinity (KM). --This statement in the abstract contradicts the observed difference in the nucleosome binding affinity.

SWI/SNF binding to nucleosomes increases the binding affinity of WT SWI/SNF for substrate over 3-times that with DNA (311 versus 98 nM)

- It has been revised to “The substrate affinity of SWI/SNF increases 3-fold with nucleosomes compared to free DNA (98 versus 311 nM) and was eliminated when the AT-hook was deleted from the complex.”

The stage-specific binding of many TFs depends on Brg1 and its AT-hook with 5,061 and 5,350 accessible sites in respectively the naïve and primed states depending on the AT-hook of Brg1 and more than 7,000 sites that remain unaltered upon deletion of the AT-hook.

- This sentence is now corrected to “There are more than 5,000 stage-specific sites in either the naïve or primed stages that depend on the AT-hook of Brg1 for TF binding as detected by ATAC-seq (Figure 5g). There are also about 7,000 sites where TF binding is retained in either the naïve or primed stage when the AT-hook is deleted (Figures S5g).”

Similarly, there is a group of 1,003 genes in DAT1 (720 genes in DAT2) that are up regulated in the primed versus the naïve state that fails to be activated in the AT mutant (Figure 7c and S7b).

- We now revised Figure 7 and modified the section accordingly.

Reviewer #4

(1) For the yeast section, it seems surprising that *snf2* delta does not affect -1/+1 nucleosomes positioning. Indeed, it has been shown that *snf2* delta (or depletion) leads to a shrinkage of the Nucleosome-Depleted Region (Rawal et al., 2018, Genes&Dev; Kubik et al., 2019, NSMB) mainly for highly expressed genes, the ones with wider NDRs. However, this is something that is not detected in this experiment.

- There are significant differences in the experimental approaches of these two studies and ours that very likely account for these differences. The Rawal et al. paper found generally there was no change in the NDR until you focused specifically on a smaller set of genes (~70) that are targeted by Gcn4. In these experiments the widening of the NFR was observed when cells were treated with SM, which is not examined in our study. In the Kubik et al. paper they used acute depletion to avoid compensating effects that occur when you create deletion different than that done in our study.

(2) Moreover, the deletion of SNF2 has a weaker phenotype than the AT-hook deletion. This does not correlate with the growth. Either there is an issue with the MNase-seq experiment or the nucleosome organization at promoters does not induce the growth phenotype. I am more in favor of problem with the MNase treatment. Did you test different concentrations of MNase? Could you provide the metagene plots not directly as a ratio over the wt condition? Instead of performing the experiment in YPD, phenotypes might be more convincing in SM or Raffinose conditions?

- Yes, different concentrations of MNase were tested to ensure samples were not over or under digested and we have no reason to think there is a problem with the MNase data. We also have now included a metagene plot which shows little change in nucleosome positioning (shifting) accompanied by a noticeable loss of nucleosomes at the -1 and -2 nucleosomes positions upon deletion of the AT-hook (Figure 4b-e). It is not surprising that there is not a tight correlation of the slow growth of the *snf2* null in YPD and its lack of effect on nucleosome occupancy and has been observed in several studies. Also it is possible that the slow growth phenotype observed with the *snf2* null is due to the dysregulation of a few genes which would not be apparent in the MNase data. We agree that doing the MNase-seq under selection

conditions as suggested would be interesting for future studies, but is beyond the scope of this paper.

(3) For the mESCs section, Figures 5C and E are very striking. However, I think it could be a control to perform the ChIP-seq of at least one naïve TF (Oct4 for example) and one primed factor (Zic3). I agree that the TOBIAS framework is validated but it is still an indirect approach. Is it possible to provide some statistics on the DNA footprint analysis?

- We thank reviewer for their suggestion but based on their own comment these additional experiments are not warranted in this already data full report. The approach is validated and should be sufficient. We have added the statistics on the DNA footprint analysis and details are mentioned in the figure legend of the revised manuscript.

(4) Statistics have to be provided for Figure 8B and C.

- Statistical analysis have been done for these two data sets and details are added in the figure legends.

(5) In general, it would be nice to have the metagene plots at the top of heatmaps. Sometimes, differences in colors are difficult to catch at first (at least, for me).

- We provide below the metagene plots and its accompanying heatmaps for your review.

Figure 5

Figure S5

Figure-S6

Figure S8

(6) A more detailed introduction would be appreciated to give more background about CRs and the results already obtained with mESCs.

-We have added some more background on mESCs and Brg1 in the Introduction section as requested.

(7) Figures 7A, C, E and G have to be better labeled with x and y axes.

-PRO-seq meta-analysis plots are labeled in the revised version of the manuscript.

(8) Line 28: "Similarly, the AT-hook is required in yeast SWI/SNF for activation of genes". The growth is tested but not the gene expression. It affects growth suggesting that...

- We have revised the sentence to read: "Similarly, growth assays suggest the AT-hook is required in yeast SWI/SNF for activation of genes involved...."

(9) Line 260: "these data demonstrate that the catalytic activity of Brg1" is again a shortcut. Since the AT-hook regulates catalytic activity, it suggests that...

- The sentence now reads as follows: "Brg1 recruitment at these sites is not changed by loss of the AT-hook and suggests regulation of the catalytic activity of Brg1 by its AT-hook is required for monomethylation of H3K4 and not for acetylation of H3K27."

(9) Line 24: "Attenuation of SWI/SNF remodeling activity by the AT-hook". If I understood well, the AT-hook enhances (not attenuates) SWI/SNF activity. If I am not wrong, I am not sure that the word "attenuation" along the publication is the good word.

- The sentence now reads "The catalytic subunit's AT-hook is required *vivo* for SWI/SNF remodeling activity in yeast and mouse embryonic stem cells."

(10) Title: can we really say "auto-regulatory"? The AT-hook is a domain that enhances remodeling, but does it regulate itself?

-The AT-hook is defined as an auto-regulatory domain because it is located within the Snf2 catalytic subunit and regulates its ATPase and nucleosome remodeling activity.

(11) I am sure that the mESCs part can be lighten and making it easier to read. I do not criticize the quality of the data which seems excellent. To give an example, Figure 5A can be switched to the Supp since it is quite close to the Figure 5B (but only for intronic and intragenic peaks). On the contrary, Figure S5F is very striking and might be moved to the main figures. In my opinion, the authors can rearrange a bit this part to make it more user-friendly.

- We have revised the figures as requested.

REVIEWER COMMENTS

Reviewer #1 (Remarks to the Author):

My concerns have been addressed. I just want to echo reviewer 4 in recommending adding metagene plots at the top of the heatmaps.

Reviewer #2 (Remarks to the Author):

In this revised manuscript the authors have addressed the majority of my previous concerns. However, the major concern is not properly addressed. I previously raised the concern that direct interactions cannot be inferred from the presented crosslinking data as these crosslinks could arise solely from close proximity. The authors have simply changed the term "interact with" to "associate with" which implies the same exact same thing. As stands the interpretation is incorrect. In order to be suitable for publication the authors must restate their results to indicate that the crosslinking data reveals only that two regions are in close proximity, or they must test for a direct interaction or association by an orthogonal technique. The authors should also avoid the use of "as previously" in the methods and provide a detailed description of all techniques.

Reviewer #3 (Remarks to the Author):

The authors make significant improvement in the revised manuscript. Yet, there are some questions remained.

- 1) I agree with review 2 that the CX-MS data cannot be used as the evidence to support the AT-hook-H3 interaction. The authors change the statement in the Results section. But in the Discussion section, they repeatedly claim the interaction, including the statement of "the AT-hook directly interacting with the H3 tail as suggested by the protein crosslinking data". The authors show that the AT-hook increases the ATPase activity in the presence of free DNA, without any requirement of the histones. This is consistent with the conventional mechanism of AT-hook in DNA-binding, and the positively charged nature of this motif. Moreover, it is known that the SnAc domain interacts with the acidic patch of H2A-H2B, as suggested by their early works (Sen, NAR 2011; Sen, MCB 2013) and clearly shown by a recent structure of the PBAF complex (Yuan, Nature 2022). It is conceivable that the AT-hook motif is tethered close to the H3 tails. Therefore, it is very likely that the CX-MS data reflect the close proximity of H3 tails and the AT-hook, instead of the real interaction. Clarification of this point is required.
- 2) The authors detected the crosslink of the H4 tail to the C-lobe of Snf2, and gave a pretty insightful discussion on the implication in the remodeling activity. The regulation of the activity of the Snf2 motor by the H4 tail was clearly described before (doi: 10.1038/nature22036). The authors seem to miss this important piece of literature.
- 3) In figure 4 legend, there is no panel J, and panel f seems to be the data of nucleosome occupancy.
- 4) The authors presented the Km values in Table 1, but the actual data are not shown, which nevertheless should be provided in the SI.

Reviewer #4 (Remarks to the Author):

Most of my concerns were convincingly answered by the authors. I fully understand that some of experiments would keep the manuscript under revision for several months for minor improvements of the data. The text was also modified according to my comments. However, there is still a need to give more background about CRs in the introduction (some sentences about the importance of CRs will be

enough but the publication starts abruptly for the moment). If the in vitro and mESCs parts are convincing, I still find the yeast part not consistent. Indeed, if there is an effect in vivo as observed with the growth (Figure 4A), it does not fit with the effect on nucleosomes (Figures 4B-I). Moreover, I still persist on the fact that -1/NDR/+1 area are more naturally sensitive to MNase digestion (due to AT-richness). Even when well controlled, these kind of differences might appear. I propose to keep the growth test but not the MNase-seq which induces some confusion and does not lead to a clear conclusion.

Once these changes made, I will support the publication of this article in Nature Communications.

REVIEWER COMMENTS

Reviewer #1 (Remarks to the Author):

My concerns have been addressed. I just want to echo reviewer 4 in recommending adding metagene plots at the top of the heatmaps.

-The metagene plots have been added to the top of the heatmaps in the main and supplementary figures as recommended.

Reviewer #2 (Remarks to the Author):

In this revised manuscript the authors have addressed the majority of my previous concerns. However, the major concern is not properly addressed. I previously raised the concern that direct interactions cannot be inferred from the presented crosslinking data as these crosslinks could arise solely from close proximity. The authors have simply changed the term “interact with” to “associate with” which implies the same exact same thing. As stands the interpretation is incorrect. In order to be suitable for publication the authors must restate their results to indicate that the crosslinking data reveals only that two regions are in close proximity, or they must test for a direct interaction or association by an orthogonal technique. The authors should also avoid the use of “as previously” in the methods and provide a detailed description of all techniques.

- We have changed the wording from “associate with” to “in close proximity” as suggested. We have also added more detailed information about the crosslinking method and analysis as indicated. The sentence stating “AT-hook does not appear to directly interact with the ATPase domain” has also been removed to avoid any conflict, although not specifically mentioned by the reviewer. We have also added additional description of CX-MS to the methods section in place of the “as previously”.

Reviewer #3 (Remarks to the Author):

The authors make significant improvement in the revised manuscript. Yet, there are some questions remained.

1) I agree with review 2 that the CX-MS data cannot be used as the evidence to support the AT-hook-H3 interaction. The authors change the statement in the Results section. But in the Discussion section, they repeatedly claim the interaction, including the statement of “the AT-hook directly interacting with the H3 tail as suggested by the protein crosslinking data”. The authors show that the AT-hook increases the ATPase activity in the presence of free DNA, without any requirement of the histones. This is consistent with the conventional mechanism of AT-hook in DNA-binding, and the positively charged nature of this motif. Moreover, it is known that the SnAc domain interacts with the acidic patch of H2A-H2B, as suggested by their early works (Sen, NAR 2011; Sen, MCB 2013) and clearly shown by a recent structure of the PBAF complex (Yuan, Nature 2022). It is conceivable that the AT-hook motif is tethered close to the H3 tails. Therefore, it is very likely that the CX-MS data reflect the close proximity of H3 tails and the AT-hook, instead of the real interaction. Clarification of this point is required.

- The sentence highlighted in yellow has been removed from the discussion. Here are the two sentences that are now in the discussion regarding the AT-hook potentially interacting with the H3-tail:

“The AT-hook has a modest secondary role in enhancing SWI/SNF affinity for nucleosomes, which could potentially be due to it interacting with the N-terminal tail of histone H3. However, given the proximity of the bromodomain to the AT-hook, the bromodomain because of its ability to bind H3K14ac could be a bridge between the AT-hook and histone H3 tail.” We think this should accommodate the concerns expressed by both reviewers 2 and 3.

2) The authors detected the crosslink of the H4 tail to the C-lobe of Snf2, and gave a pretty insightful discussion on the implication in the remodeling activity. The regulation of the activity of the Snf2 motor by the H4 tail was clearly described before (doi: 10.1038/nature22036). The authors seem to miss this important piece of literature.

- We have added this reference as suggested of the structure of Snf2 alone bound to nucleosomes and compare it to that obtained with the complete yeast SWI/SNF complex bound to nucleosomes where the H4 tail is not observed.

3) In figure 4 legend, there is no panel J, and panel f seems to be the data of nucleosome occupancy.

-We have corrected the mistake as pointed out.

4) The authors presented the Km values in Table 1, but the actual data are not shown, which nevertheless should be provided in the SI.

- The actual data for the Km values is shown in Figure 2a-d for each concentration of ATP used in these assays. The raw data is include in the source file.

Reviewer #4 (Remarks to the Author):

Most of my concerns were convincingly answered by the authors. I fully understand that some of experiments would keep the manuscript under revision for several months for minor improvements of the data. The text was also modified according to my comments. However, there is still a need to give more background about CRs in the introduction (some sentences about the importance of CRs will be enough but the publication starts abruptly for the moment). If the in vitro and mESCs parts are convincing, I still find the yeast part not consistent. Indeed, if there is an effect in vivo as observed with the growth (Figure 4A), it does not fit with the effect on nucleosomes (Figures 4B-I). Moreover, I still persist on the fact that -1/NDR/+1 area are more naturally sensitive to MNase digestion (due to AT-richness). Even when well controlled, these kind of differences might appear. I propose to keep the growth test but not the MNase-seq which induces some confusion and does not lead to a clear conclusion.

Once these changes made, I will support the publication of this article in Nature Communications.

- We thank the reviewer for the suggestions to improve our manuscript. We have added sentences about the importance of CRs in the introduction. The MNase data from figure 4 has been removed due to the concerns of this reviewer.